



# High-resolution mapping of urban NO₂ concentrations using Retina v2: a case study on data assimilation of surface and satellite observations in Madrid

Bas Mijling[1], Henk Eskes[1], Sascha Hofmann[2], Pau Moreno[2], David García Falin[3], María Encarnación de Vega Pastor[3]

[1]Royal Netherlands Meteorological Institute, 3730 AE De Bilt, the Netherlands
[2]Lobelia Earth, 08005, Barcelona, Spain
[3]Government Area of Urban Planning, Environment and Mobility, Madrid City Council, Madrid, Spain

*Correspondence to*: Bas Mijling (bas.mijling@knmi.nl)

**Abstract.** Urban air pollution poses a significant health risk, with over half the global population living in cities where air quality often exceeds World Health Organization (WHO) guidelines. A comprehensive understanding of local pollution levels is essential for addressing this issue. Recent advancements in low-cost sensors and satellite instruments offer cost-efficient complements to reference stations but integrating these diverse data sources in useful monitoring tools is not straightforward. This study presents the updated Retina v2 algorithm, which generates high-resolution urban air pollution maps by assimilating heterogeneous measurements into a portable urban dispersion model. Tested for NO₂ concentrations in Madrid during March 2019, it shows improved speed and accuracy over its predecessor, with the ability to incorporate satellite data. Retina v2 balances performance with modest computational demands, delivering similar or better results compared to complex dispersion models and machine learning approaches requiring extensive datasets. Using only TROPOMI satellite data, citywide NO₂ simulations show an RMSE of 19.3 µg/m³, with better results when hourly in-situ measurements were included. Relying on data of a single ground station can introduce biases, which can be mitigated by incorporating satellite data or multiple ground stations. Including more stations improves accuracy, with 24 stations yielding a correlation of 0.90 and an RMSE of 13.0 µg/m³. The benefit of TROPOMI diminishes when data from five or more ground stations is available, but it remains valuable for many cities which have limited monitoring networks.

## 1 Introduction

More than half of the world's population lives in cities, where most people breath air that exceeds the World Health Organization's (WHO) air quality guidelines (WHO, 2021). Elevated levels of nitrogen dioxide (NO₂), primarily from urban traffic and residential emissions, significantly contribute to this health issue. NO₂ is linked to respiratory diseases, particularly asthma, leading to respiratory symptoms (such as coughing or difficulty breathing), hospital admissions and visits to emergency rooms. According to the WHO air quality database (WHO, 2023), 77% of the population in the 4,000



assessed towns and cities are exposed to mean annual $NO_2$ levels above the recommended limit of 10 µg/m³. This figure rises to 90% in cities in low- and middle-income countries, paralleling large-scale urbanisation and economic development.

Addressing urban air pollution requires a detailed understanding of local pollution levels. This is best achieved with a dense network of reference stations, as traffic patterns and urban design can cause strong gradients in air pollution (Cummings et al., 2022). However, the high costs of installing and maintaining such networks often leave cities, especially in low- and

middle-income countries, without adequate monitoring infrastructure.

In recent years, alternative air quality measurements from low-cost sensors and satellite instruments have become available. The monitoring of urban air quality will greatly benefit from incorporating these complementary measurements (WMO, 2024). However, integrating different data sources in a transparent manner is challenging because they differ in sampling frequencies and spatial representativeness. While low-cost air quality sensors can provide detailed spatial observations in

urban areas, they often come with significant uncertainties (Snyder et al., 2013). As satellites observe air pollution of the entire troposphere, the relationship between column concentrations and surface-level concentrations must be resolved first. Also, polar-orbiting satellites pass over the same area only once per day, missing a substantial part of the diurnal cycle (Boersma et al., 2009).

Air quality models are therefore essential to create maps from measurements, as they not only fill in the unsampled areas and

times, but also (in the more advanced data fusion methods) consider the different spatial representativity and accuracy of the measurements, and —for satellite measurements— the height distribution in column measurements.

Modelling at the urban scale can be done by Land Use Regression (LUR) models (Hoek et al., 2008), which solve statistical relations between surface concentrations and geographic data. They are commonly used in exposure studies, providing maps at high spatial resolution but lacking a time component. Another approach is to downscale the output of regional chemical

transport models to high-resolution sub-grid concentrations (e.g. Denby et al., 2020, Kim et al., 2018). Like LUR models, Gaussian plume models (e.g. listed in Kakosimos et al., 2010) are widely used in urban settings due to their low computational demands. Based on an analytical solution to pollutant transport equations and a detailed emission inventory, they can calculate hourly concentrations of air pollutants at street-level under given meteorological conditions.

Better simulation results are obtained when in-situ measurements are spatially assimilated in modelled concentration fields

using kriging techniques (Schneider et al., 2017, Criado et al., 2023) or optimal interpolation (Tilloy et al., 2013, Mijling 2020). This significantly reduces both local biases and background biases.

The TROPOspheric Monitoring Instrument (TROPOMI) is a nadir-viewing imaging spectrometer aboard ESA's Sentinel-5P satellite. Since May 2018, TROPOMI has provided global observations of air quality from space with an unprecedented spatial resolution (5.6 x 3.6 km$^2$ at nadir view since 6 August 2019). This resolution offers coarse information on spatial

patterns of air pollution within urban environments. For instance, tropospheric $NO_2$ column measurements of TROPOMI



have been used to estimate NOx emissions in Paris (Lorente et al., 2019), to predict daily surface $NO_2$ concentrations in Mexico City (He et al., 2023), and to detect spatiotemporal variations of $NO_2$ in Madrid (Morillas et al., 2024).

TROPOMI observations are used by Kim et al. (2021) to create hourly $NO_2$ maps for Switzerland and northern Italy on a 100m resolution, in a combination with reference measurements, geographical and meteorological data. Fu et al. (2023) also
add low-cost sensor data for hourly mapping in Tangshan, China. Both studies show that satellite data can contribute significantly to surface $NO_2$ mapping, despite its coarse resolution and the fact that it is only available once a day under near cloud-free conditions.

Recurrent complication in urban air quality modelling is the need of an up-to-date emission inventory at a high resolution, and realistic estimations of the regional background concentrations. Also, many urban data fusion applications depend on
machine learning (e.g. Kim et al., 2021, He et al., 2023) or a detailed local air quality model (e.g. Schneider et al., 2017, Criado et al., 2023). This complicates portability to other cities where the required input data might not be available.

The Retina algorithm (Mijling 2020) provides a physics-based and portable approach. It has been developed specifically for observation-based high-resolution modelling of urban air pollution using heterogeneous air quality measurements (i.e. of different accuracy and origin). Central in Retina is the open-source AERMOD dispersion model (Cimorelli et al., 2004),
developed by the American Meteorological Society (AMS) and United States Environmental Protection Agency (EPA). The model is driven by meteorology and local emissions constructed from proxy data. Observations are used for emission optimization and spatial concentration assimilation. It generates hourly maps of air pollutant concentrations at street-level.

Retina v2, described in this paper, has undergone significant updates to enhance its speed and accuracy. It is faster and uses less computational resources by using AERMOD only for dispersion kernel calculations (Sect. 2.3.1). The estimation of
background concentrations has been improved (Sect. 2.2.1). The $NO_2$/NOx ratios are estimated more accurately by replacing the Ozone Limiting Method with a non-linear regression method (Sect. 2.3.2). The estimation of the sectoral emission factors is better stabilised by implementation of a Kalman filter (Sect. 2.3.4). The spatial assimilation of concentration measurements is improved by including time-dependent dispersion characteristics in the model error covariances (Sect. 2.3.5). Most notably, for the CitySatAir project (part of ESA's EO Science for Society program) we extended the
algorithm with an additional module to incorporate tropospheric column concentrations of $NO_2$ measured with TROPOMI (Sect. 2.3.3).

## 2 Method

The added value of $NO_2$ column measurements from space is evaluated through their application in Madrid, Spain, for the period of March 2019. The city's extensive network of $NO_2$ reference stations allows for the exploration of different
measurement configurations, including scenarios with and without TROPOMI observations.

The municipality of Madrid extends over an area of about 40 x 43 $km^2$, with a population of approximately 3.4 million people. Urban $NO_2$ pollution levels are amongst the highest in Europe, regularly exceeding the air quality guidelines set by



the WHO (WHO, 2021), which recommend limits of 10 μg/m$^3$ for annual averages and 25 μg/m$^3$ for daily averages. The city of Madrid ranks first in Europe for mortality linked to NO$_2$ pollution, according to a recent health impact study by ISGlobal, which analysed nearly 1000 European cities (Khomenko et al. 2022). Traffic and residential emissions are mainly responsible for the high surface concentrations found in the city, as Madrid has no heavy industry or other important NO$_2$ hot spots in its immediate vicinity.

## 2.1 NO$_2$ observations

### 2.1.1 Reference network

Common equipment to perform reference measurements of NO$_2$ include the Teledyne API 200E and a Thermo Electron 42i NO/NOx analyser, both based on chemiluminescence. A catalytic–reactive converter converts NO$_2$ in the sample gas to NO, which, along with the NO present in the sample, is reported as NOx. NO$_2$ is then calculated as the difference between NOx and NO. We use an accuracy of 4% of the NO$_2$ measurements (GGD, 2014). Note that this might be an underestimation for locations downwind of source areas, as the molybdenum converter also reduces other reactive nitrogen species such as PAN and HNO$_3$ (especially found in aged plumes) to NO, introducing a positive bias in the NO$_2$ measurements (Steinbacher et al., 2007).

Madrid has an extensive network of 24 reference sites measuring hourly NO$_2$ concentrations (see Fig. 1), from which 9 qualify as street stations, 12 as urban background stations, and 3 as suburban background stations. Hourly measurements of the network are published in near real-time as open data at the Madrid Open Data Portal (https://datos.madrid.es). Lower concentrations of NO$_2$ are found in summertime, due to favourable atmospheric conditions (e.g. higher boundary layer height) and less emissions during the holiday period. Highest concentrations are found in wintertime, when monthly averaged values are well above 40 μg/m$^3$ at roadside and urban background sites. The network-wide average in March 2019 (36.2 μg/m$^3$) closely reflects the annual average (34.5 μg/m$^3$).





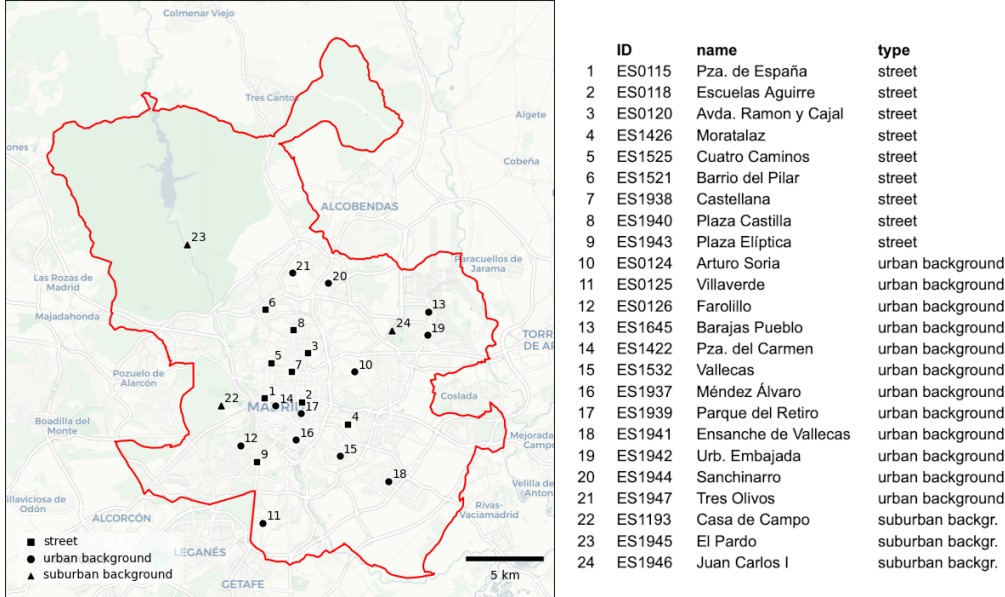

**Figure 1: Reference network for NO₂ measurements in Madrid. The red line indicates the municipality border. Basemap source: © OpenStreetMap contributors © Carto, 2024. Distributed under a Creative Commons BY-SA License.**

### 2.1.2 TROPOMI retrievals

TROPOMI is a nadir-viewing imaging spectrometer aboard ESA's Sentinel-5P satellite. Since May 2018, TROPOMI has provided global observations of air quality from space with an unprecedented spatial resolution ($5.6 \times 3.6$ km$^2$ at nadir view since 6 August 2019), enabling it to offer coarse information on spatial patterns and gradients of air pollution within urban environments.

Being in a sun-synchronous orbit at 824 km altitude, Sentinel-S5P overpasses Madrid daily around 13:00 UTC. At times the urban area is sampled from two adjacent orbits, typically around 12:30 and 14:10 UTC. At every overpass the retrieval footprints are located differently, sampling different parts of the urban area.

From the radiance and irradiance spectra the NO₂ slant column density can be derived, which is divided into a stratospheric and tropospheric part. By applying an appropriate air mass factor the tropospheric slant column is converted to a tropospheric vertical column density and its accompanying averaging kernel. We use the latest reprocessed product for TROPOMI tropospheric NO₂ columns, version 2.4 (Eskes et al., 2022; van Geffen et al., 2022), which implements a new surface albedo climatology derived from TROPOMI observations, including the viewing-angle dependence of the scattering at the surface.

Although it is recommended that for straightforward application only retrievals with a quality value ≥0.75 should be used (i.e. valid retrievals with cloud radiance fractions below 50%), this criterion is relaxed to ≥0.5 (i.e. valid retrievals, including under cloudy conditions) as the averaging kernel is carefully applied (see Van Geffen et al., 2022). Also, only footprints



which cover the studied domain by more than 50% are used. In this way, there are on average 14.2 valid retrievals found per day in March 2019.

## 2.2 Model input data

### 2.2.1 Background concentrations

An important fraction of the air pollution is transported from upwind regions (see e.g. Harrison, 2018). Using realistic
background concentrations of $NO_2$ is therefore crucial as the dispersion simulation only accounts for locally generated $NO_2$ within the domain.

Here we use hourly data from the European air quality ensemble from the Copernicus Atmosphere Monitoring Service (CAMS) (Marécal et al., 2015, METEO-FRANCE et al., 2020), which for 2019 data consist of 9 state-of-the-art numerical air quality models. More specifically, the ensemble median of the validated reanalysis is used, a data product for which each
ensemble member assimilated validated hourly observations of air pollutants as reported to the European Environment Agency. The CAMS regional ensemble covers Europe at a resolution of 0.1 x 0.1 degree. Note that the coarseness of the data product makes it unsuitable for monitoring urban air quality in detail, as the strong concentration gradients found around strong sources will be averaged out, leading to an underestimation of $NO_2$ concentrations, see Fig. 2.

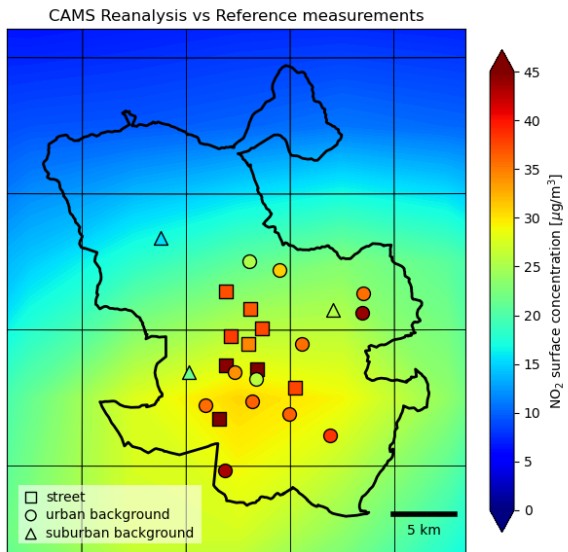

**Figure 2: $NO_2$ surface concentrations in Madrid, March 2019, from bilinear interpolation of the average values of the CAMS reanalysis. CAMS grid cells are shown in thin black lines. Also shown are the mean concentrations measured by the reference network.**

To avoid double counting of locally produced $NO_2$ concentrations, we take for each hour a weighted average of the CAMS concentrations found along the municipal perimeter. Let $b_C(\mathbf{x})$ represent the interpolated CAMS concentration at location $\mathbf{x}$,
$\mathbf{e}_v$ the unit vector along the wind direction, $\mathbf{n}$ the normal vector on the perimeter (pointing outwards), and $\mathcal{L}$ the part of the





perimeter where $\mathbf{e}_v \cdot \mathbf{n} < 0$ (i.e. where background pollution is flowing into the municipal domain). Then the weighted average for this hour is calculated from the line integral

$$b = \frac{1}{W} \int_{\mathcal{L}} b_C(\mathbf{x}) \mathbf{e}_v \cdot \mathbf{n} \, dl \text{ , with } W = \int_{\mathcal{L}} \mathbf{e}_v \cdot \mathbf{n} \, dl \text{ ,} \tag{1}$$

where $dl$ represents the elementary arc length along the perimeter. The resulting background value $b$ is taken homogeneously for the entire domain, thus avoiding the use of more advanced advection schemes. The background concentration of ozone, needed in the calculation of the $NO_2/NOx$ ratio is calculation from CAMS ozone fields in the same manner.

### 2.2.2 Meteorology

Meteorology is an important ingredient for dispersion modelling, as it determines how the air pollutant is transported horizontally and vertically. AERMET, the meteorological preprocessor of AERMOD, requires both surface meteorological data (cloud cover, temperature, humidity, dew point temperature, pressure, precipitation, wind speed and direction) and upper air meteorological data (temperature, humidity, wind speed and direction in vertical layers). The wind speed and direction are also used to determine the influx of background concentrations and the main axes of the model error covariance (Sect. 0).

We use the collection of short-range forecasts (issued at 0 UTC and 12 UTC) from the archive of the European Centre for Medium-Range Weather Forecasts (ECMWF) at the supercomputing facility in Bologna. It is retrieved as 3-hourly output at a 0.05-degree spatial resolution. Hourly meteorological fields are obtained by temporal interpolation and then interpolated to a representative location central in the Retina domain.

### 2.2.3 Emission proxies

An important unknown when modelling street-level urban air pollution is the location and strength of the urban emissions. For many cities this information is either unavailable or outdated. By describing traffic and residential emissions with proxies taken from open data sets (see Fig. 3) we enable a versatile model setup which can be applied easily to other cities. The domain boundary is extended outward by 1500 m to account for contributions from sources close to the municipal border. Other sectoral emissions, e.g. from industry, will be accounted for indirectly in either an increased background field or in additional residential emissions.



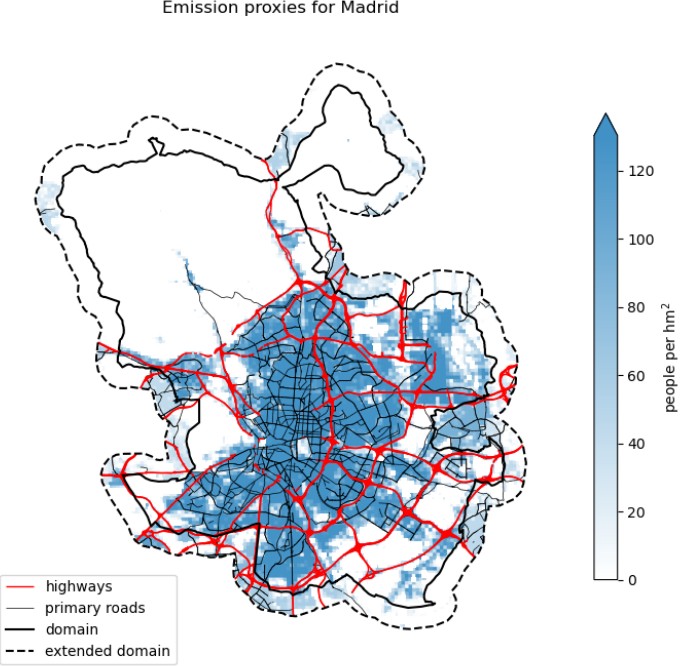

**Figure 3: Proxies used for the estimation of urban emissions. The Retina algorithm distinguishes between highways, primary roads, and residential emissions. Nearby emissions outside the municipal border are also considered.**

Road location and road type classification is taken from OpenStreetMap (OSM). All road segments labelled "motorway" or "trunk" are linked to the highway class. All "primary", "secondary" and "tertiary" segments are linked to the primary road class.

The Madrid Open Data Portal publishes vehicle counts at approximately 4000 locations, mainly from inductive loop sensors at traffic junctions. The data files contain vehicle counts in 15-minute bins, which are aggregated into one-hour bins to align with the temporal resolution of the simulations. For each location, a monthly averaged traffic volume cycle is calculated, separated by hour of day and day of week.

Between counting locations, the traffic flow is estimated by spatial interpolation using inverse-distance weighting. The interpolation is done separately for vehicle counts at highways and primary road networks, as they have incomparable traffic volumes.

Population density is a good proxy for residential emissions from activities such as heating and cooking. We use the population densities from the Global Human Settlement project (Freire et al., 2016) which are provided on a 250 m resolution. To reflect the observation that per capita residential emissions decrease when people live closer to each other (e.g. Makido et al., 2012), the emission fluxes are scaled proportionally to the square root of the population density.



The proxy data $P_s$ for sector $s$ (here traffic and residential) are gridded on a high-resolution 10 m grid to enable fast application of the dispersion kernel (see 0). Sectoral NOx emissions for a grid cell indexed with $(i,j)$ are calculated by

applying the emission factors $f_s$ to the proxy data:

$$E_s(i,j) = f_s P_s(i,j) \qquad (2)$$

Emissions change over the day. For traffic emissions this is described by a time dependency in the proxy data. The diurnal cycle in residential emissions, however, is described by 24 different emission factors (each for one hour), as its proxy data are constant in time.

**2.3 The revised Retina algorithm**

Retina uses past observations for emission optimisation (minimizing the general model bias) and current observations for spatial concentration assimilation (reducing local model biases).

In the emission optimisation step (represented in Fig. 4) the algorithm fits emissions factors for the proxy emissions that best match the spatio-temporal concentration patterns observed by the reference network. The optimisation is repeated every 24

hours (see Sect. 0), using the emission factors and covariances of the previous estimation as *a priori*. This approach avoids the need of detailed knowledge of vehicle fleet composition and solves mismatches between theoretical and real-world emissions. It also compensates for model biases resulting from incorrect chemistry (e.g. lifetime) and unaccounted (seasonal) emission cycles.

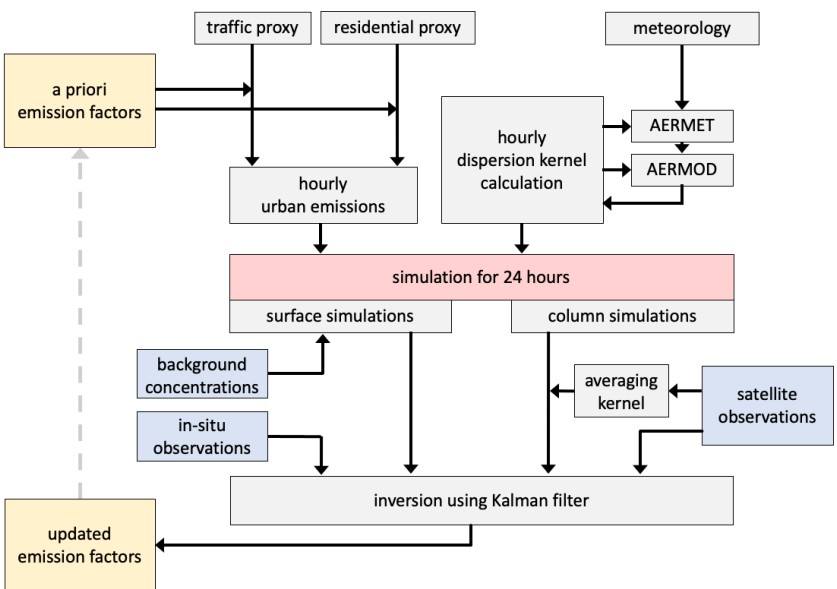

**Figure 4: Schematic representation of Retina's emission optimisation workflow. Starting with *a priori* emission factors, the AERMOD dispersion model simulates NO₂ concentrations at the observation locations for a 24-hour period. The Kalman filter**



**infers from the difference between observation and simulation the best update for the emission factors. These values are passed to the next analysis period.**

With the most recent estimation of emission factors, Retina simulates the surface concentrations for a specific hour at all
62,266 locations of a non-regular, road-following mesh (see e.g. Lefebvre et al., 2011).

Next, in-situ observations are spatially assimilated in the simulated concentration field using optimal interpolation (Daley, 1991), see Fig. 5. This technique allows for the assimilation of surface measurements with different error margins. At the observation locations, model values are corrected towards the observations. In the surrounding areas, the balance between the model and observation errors determine how the simulation is adjusted (see Sect. 2.3.5).

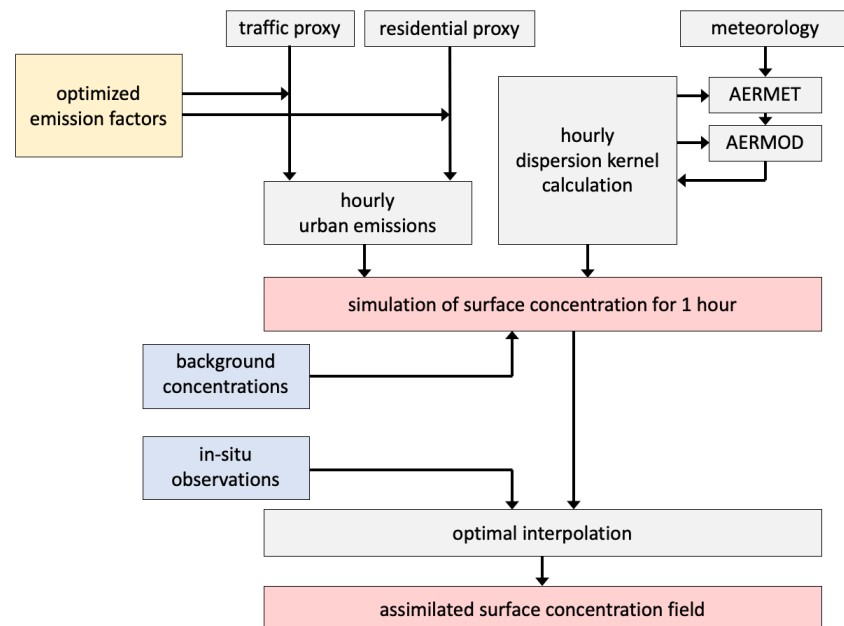


**Figure 5: Schematic representation of Retina's hourly simulation and assimilation workflow. The dispersion model uses optimised emissions and background concentrations to simulate surface concentrations. In-situ observations (if available) are assimilated in an optimal interpolation scheme.**

### 2.3.1 Dispersion kernel

Due to the large number of emission sources, using a straightforward AERMOD configuration can result in long calculation times, especially if simulations must be performed at several vertical levels to recreate the tropospheric columns. Instead, we adopt the approach suggested by Masey et al. (2018), using AERMOD exclusively to calculate the dispersion kernel. This is the dispersion of a unit NOx emission for a specific hour under given meteorological conditions (e.g., wind speed and direction, atmospheric stability, and boundary layer height).

Removal processes of NOx are modelled with an exponential decay, which is included in the kernel calculation. In urban settings, the typical lifetime of NOx is on the order of a few hours (Beirle et al., 2011), and changes significantly when plumes travel from source areas and mix with clean air (Krol et al., 2024). However, as emitted NOx has a relatively short




residence time in urban areas, the specific value of its lifetime is not very critical. We use a heuristic value of 2 hours throughout this study.

Dispersion kernels are computed for all emission release heights of the emissions and all receptor heights. Sectoral release heights are 0.5 m for traffic and 10 m for residential emissions. These sector-specific dispersion kernels $K_s$ are then gridded onto the regular high-resolution grid, aligning with the emission grid $E$. Examples of dispersion kernels are illustrated in Fig. 6.

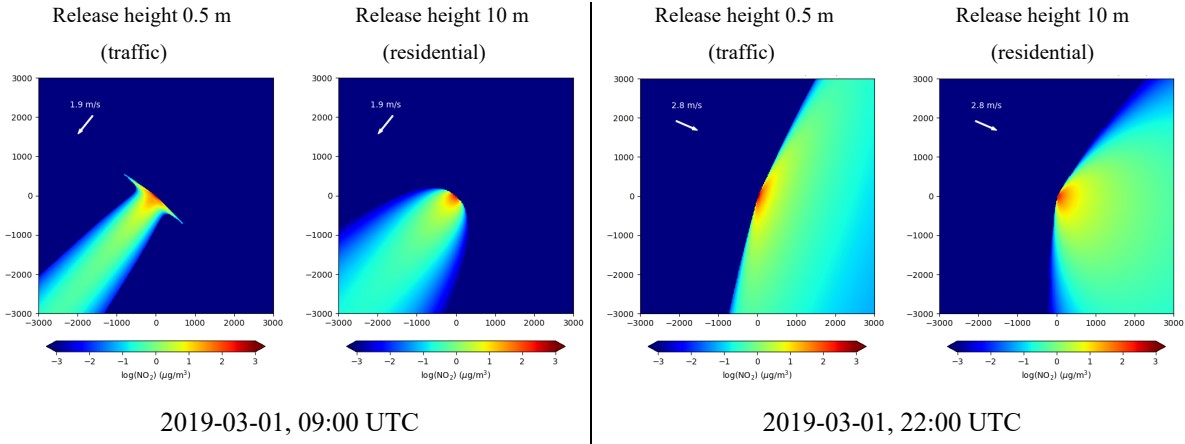

**Figure 6: Examples of dispersion kernels at different release heights and winds (i.e. different atmospheric stability) for surface concentration calculation. The receptor height of the kernels is at 1,5 m. Distances are given in meters.**

The NOx concentration $c^{\text{NOx}}$ (in NO$_2$ mass equivalent per volume) for a receptor located in grid cell $(i,j)$ can now be calculated as a superposition of all dispersed emissions, for all contributing emission sectors $s$

$$c^{\text{NOx}}(i,j) = \sum_s \sum_{i',j'} K_s(i-i', j-j') E_s(i',j') \tag{3}$$

This can be interpreted as an element-wise matrix multiplication (i.e., the Hadamard product) of the mirrored dispersion kernel with its origin at $(i,j)$ with the entire emission grid. In first order, the NOx concentrations depend linearly on contributing sources; an NO$_2$/NOx ratio is applied afterwards (see Sect. 0). Transport over longer distances must be accounted for to prevent underestimation: although a single grid cell at a long upwind distance may contribute only a small

amount, the cumulative effect from numerous such grid cells becomes significant. In our algorithm we consider contributions up to 30 km to account for transport effects in the entire simulation domain. To ensure computational efficiency, contributions at longer distances are computed at lower resolutions, as emission source locations become less critical with increasing distance.



### 2.3.2 Surface concentration simulation

We calculate $NO_2$ concentrations from NOx concentrations using a time and location dependent $NO_2$/NOx ratio. The dynamic equilibrium between NO and $NO_2$ is affected by temperature, available ozone (which generates $NO_2$ from NO), and solar radiation (which generates NO from $NO_2$). The local $NO_2$/NOx ratio is hourly estimated using parameters available during simulation:

- the local simulated NOx concentration.

- the background $O_3$ concentration, taken from the regional CAMS ensemble.

- the background $NO_2$ concentration, also taken from the regional CAMS ensemble.

- the temperature, as a measure of reaction speed for conversion NO to $NO_2$.

- the Solar Elevation Angle (SEA), as a measure of radiation available for photolysis of $NO_2$.

As the dependence on these parameters is non-linear, we train an extreme gradient boosting (XGBoost) model (Chen and
Guestrin, 2016) to estimate $NO_2$ ratios at simulation time (see Sect. S1 in the Supplement).

From the NOx concentration $c^{NOx}$ calculated by Eq. 3, the $NO_2$/NOx ratio $r$, and the background concentration $b$ (from Eq. 1), we calculate the $NO_2$ surface concentration $c$ as

$$c(i,j) = b + r(i,j)c^{NOx}(i,j) \tag{4}$$

### 2.3.3 Column concentration simulation

For column concentration simulations, which are necessary for comparison against satellite observations, we use the same approach as for surface simulations but with different settings (see Table 1).

**Table 1: Overview of simulation settings**

| Surface simulation | Tropospheric column simulation |
| --- | --- |
| Surface level only (1.5 m) | 9 horizontal levels: 0, 1.5, 50, 250, 500, 1000, 2000, 3000, 5000 m |
| 10 m horizontal resolution | 250 m horizontal resolution |
| $NO_2$ ratio from XGBoost model | $NO_2$ ratio from CAMS |
| $NO_2$ background concentration from CAMS | $NO_2$ background column from fit |

Given the large footprints of the satellite observations relative to the urban domain, it is crucial to maximize the information content from single $NO_2$ retrievals. Gridding and averaging to a model grid, or clustering orbits in time, would result in



valuable information loss. Instead, we project simulation results to individual footprints in individual orbits, following these steps:

1. Modelling of $NO_X$ concentrations at high resolution at 9 different heights. We use the vertical grid definition from the CAMS regional ensemble from surface to 5 km.

2. Spatially aggregating the simulated values to individual satellite footprints. The coarser resolution in the horizontal (250 m) is justified as the satellite footprints are in the order of kilometres.

3. Temporal interpolation of simulation values to the exact satellite overpass time.

4. Applying the averaging kernel associated with the retrieval, after conversion of the concentration profile to partial columns matching the kernel's levels. This reduces errors resulting from profile assumptions in the retrieval method (Eskes and Boersma, 2003).

Finding a realistic $NO_2/NO_x$ ratio for column simulation is not straightforward, as the chemical equilibrium changes with height. Close to sources, lower layers exhibit smaller ratios at low temperatures (when conversion from primarily emitted NO is not established yet), while higher layers show reduced $NO_2$ due to stronger photodissociation from solar irradiance. An intricate chemical analysis is avoided by taking the ratio $R$ from columns of the CAMS regional ensemble (see S2 in the Supplement). For Madrid, $R$ fluctuates between 0.4 and 0.8, with the lowest values found in winter. Note that $NO_2/NO_x$ ratios in columns are generally higher than surface ratios due to increased ozone availability.

The column simulation includes only local contributions, excluding the background column (i.e., concentrations from emissions outside the domain and the free tropospheric column above the boundary layer). To simplify matters, we do not simulate this background column. Instead, the background column concentration is determined for each overpass by fitting the simulated column concentrations to the observations.

Let $C_{kls}^{NOx}$ be the simulation of the partial NOx column concentration for the footprint of retrieval $k$, for atmospheric layer $l$, and for emitting sector $s$. $C_{kls}^{NOx}$ is calculated from dispersed proxy data following step 1 to 3. Let $A_{kl}$ be the averaging kernel element for layer $l$ and retrieval $k$. With the ratio $R$ known from CAMS and the background column $B$ fitted, we can write for the simulated tropospheric $NO_2$ column $C$:

$$C_k = B + R_k \sum_l A_{kl} \sum_s C_{kls}^{NOx} \tag{5}$$

Figure 7 demonstrates that this approach effectively simulates the location, shape, and strength of the pollution plume over the city. The $y$-intercept in the scatter plots represents the estimated background concentration $B$, which is determined using ordinary least squares regression, using the reciprocal of the retrieval error as error weights.







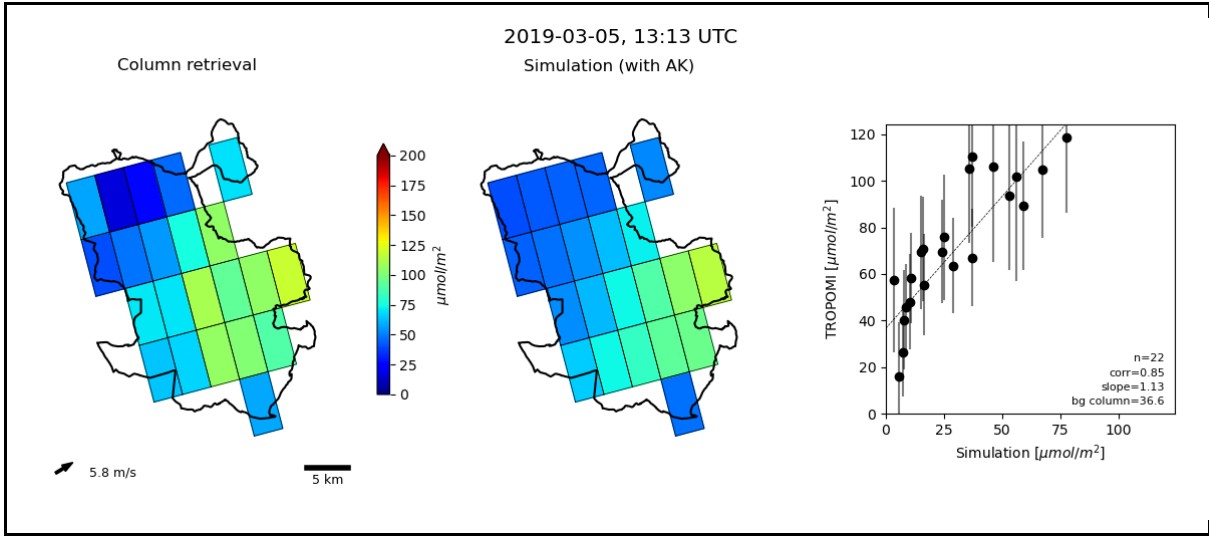


**Figure 7: Three examples of tropospheric NO₂ columns over Madrid under different wind speeds and directions (indicated by the black arrows). Left panels show retrievals from TROPOMI in single overpasses. Middle panels show corresponding column simulations (including background columns) by Retina, where emission factors have been optimised before against surface concentrations. The background columns are estimated from the linear fit between simulations from local emissions and**
**observations (right panels).**

### 2.3.4 Emission optimisation: estimating emission factors

In the dispersion modelling described above, the sector-specific emission factors $(f_s)$ remain unknown. These can be estimated from the observations. Rewriting Eqs. (2) to (5), we can express simulations of both NO₂ surface concentrations at location $i$ and tropospheric NO₂ column concentrations at footprint $k$ as a linear combination of $f_s$:

$$c_i = b + r_i \sum_s \alpha_{is} f_s \ \text{ and } \ C_k = B + R_k \sum_s \beta_{ks} f_s \qquad (6)$$

Here, $\alpha_{is}$ and $\beta_{ks}$ are calculated from the dispersion of sectoral proxy data by the dispersion kernels, $b$ and $B$ are the background concentration at the surface and the background column concentration respectively, and $r$ and $R$ the corresponding NO₂/NOx ratios.

Finding the emission factors from observations is an ill-posed inverse problem, which we regulate here with a Kalman filter.
The technical description of the estimation is described in Sect. S3 of the Supplement. For in-situ observations, the state vector consists of 25 unknown $f_s$ values: one emission factor for traffic, and 24 elements describing the diurnal cycle of the residential emissions. This is different when using TROPOMI observations only, as they only provide information around overpass time. In this case, we use an *a priori* diurnal cycle for residential emissions (see Sect. S4). The state vector is then reduced to two unknowns: one emission factor for traffic and one factor that scales the *a priori* residential diurnal cycle.

By carefully selecting the covariances in the filter we optimise the response time without introducing too much noise from measurement and model errors. Starting with an arbitrary state vector, a spin-up time of at least one month (i.e. ~30



optimisation iterations) is needed. Figure 8 shows an example how satellite observations of a single orbit over Madrid are used for an updated estimation of the emission factors.

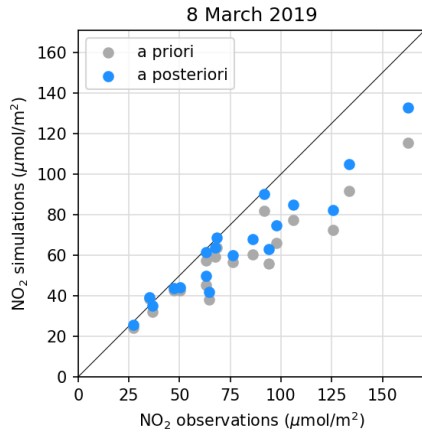

**Figure 8: Example of the Kalman filter applied to TROPOMI observations in a single orbit over Madrid on March 8, 2019. The grey dots represent the simulated tropospheric column concentration values, which show a clear underestimation for that day. After applying the Kalman filter the emission factors are updated, resulting in new simulations (blue dots) that are closer to the $x = y$ line. The updated emission factors are used as *a priori* in the emission optimisation of the following day.**

### 2.3.5 Spatial assimilation of surface concentrations

Retina uses the optimised emissions to simulate hourly surface concentration fields, which serves as *a priori* for the spatial assimilation of in-situ measurements. This assimilation process corrects local model errors from e.g. incorrect proxy data or inhomogeneities in the background field. With vector $\mathbf{x_f}$ representing the simulated surface concentration field (i.e. the *forecast)*, and vector $\mathbf{z}$ containing the in-situ observations, the statistical interpolation can be written as:

$$\mathbf{x_a} = \mathbf{x_f} + \mathbf{K}(\mathbf{z} - H(\mathbf{x_f})) \tag{7}$$

$$\mathbf{K} = \mathbf{P^f H^T}(\mathbf{H P^f H^T} + \mathbf{R}) \tag{8}$$

The update of the forecasted field ($\mathbf{x_a} = \mathbf{x_f}$) depends on the difference between the observations $\mathbf{z}$ and the collocated simulations $H(\mathbf{x_f})$. Matrix $\mathbf{H}$ maps the simulations to the observation locations, and $\mathbf{R}$ contains the observation covariances, as in Mijling (2020). The update is determined by the Kalman gain matrix $\mathbf{K}$ which balances between the observation error and the model error. Figure 9 illustrates this spatial assimilation cycle.

The model error covariance matrix, $\mathbf{P^f}$, represents the spatial extent of model errors: an error at the observation location implies errors in nearby areas. This covariance is approximated by accounting for the spatial representativity of observations, which varies between street and background locations, and by incorporating hourly changes in atmospheric dispersion, writing:





$$\text{cov}(\mathbf{x}_1, \mathbf{x}_2, t) = \sigma_1 \rho_A(\mathbf{x}_1, \mathbf{x}_2) \rho_B(\mathbf{x}_1 - \mathbf{x}_2, t) \sigma_2 \ , \tag{9}$$

where $\sigma_1$ and $\sigma_2$ are the model errors at location $\mathbf{x}_1$ and $\mathbf{x}_2$. $\rho_A$ represents the correlation between modelled time series at these locations (which is here calculated from all simulations for March 2019). Correlations between traffic locations and background locations will be lower than between similar locations, therefore $\rho_A$ reduces the impact of an observation on areas it is not representative of (see Fig. S6).

New in Retina v2 is the inclusion of $\rho_B$, representing the spatial correlation due to the time-dependence in pollutant

dispersion. We want to describe this in terms of the two-dimensional dispersion kernel introduced in Sect. 0. As demonstrated in Sect. S5 of the Supplement, $\rho_B$ can be calculated by multiplying the 2D kernel with a copy of itself, shifted by the distance $\mathbf{x}_1 - \mathbf{x}_2$. This results in a dispersion correlation field which is symmetric along the downwind and crosswind axes. The correlation at a distance $d$ along each main axis can be approximated with

$$\rho_B(d) \approx (1 + |d/L|)^{0.75} \exp(-|d/L|^{0.75}) \ , \tag{10}$$

with $L$ the fitted correlation length for the considered hour. Typically, the correlation lengths are larger at night and shorter during the day. The exponent 0.75, determined heuristically, provides representative high correlations around the measurement location and best captures the decaying tail.




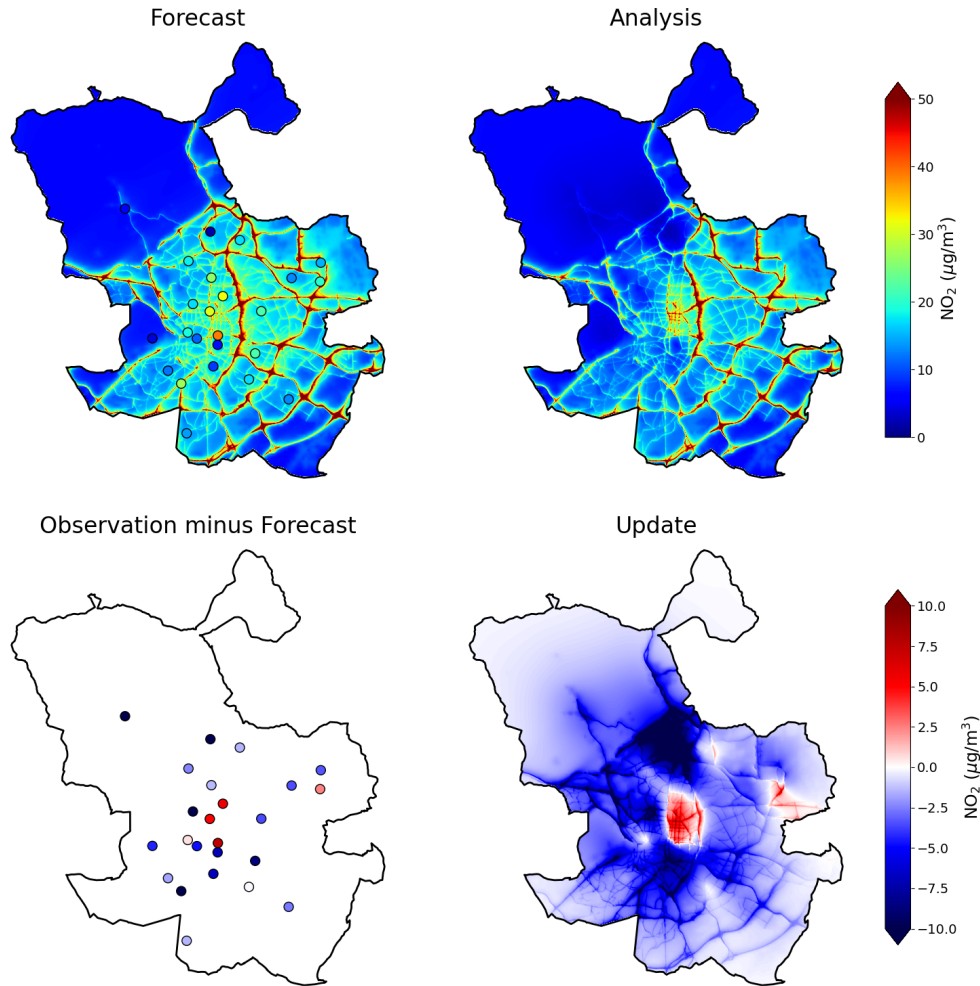

**Figure 9: Spatial assimilation for 7 March 2019, 13:00. Starting from the top-left panel and moving counter-clockwise: in-situ observations and the forecasted field based on optimised emission factors; difference between observations and simulation; concentration update using optimal interpolation; a posteriori field with assimilated observations.**

## 3 Results

### 3.1 Model accuracy with TROPOMI-only data

As explained in Sect. 2.3.3 and 2.3.4, retrievals in individual overpasses of TROPOMI can be used to optimise the emissions for the dispersion model. We evaluate the results when optimisation is done at 24-hour intervals, with a two-month spin-up time to ensure convergence from *a priori* values. The space-derived emission factors are used to simulate hourly surface concentrations at a high resolution, as shown in Fig. 10.





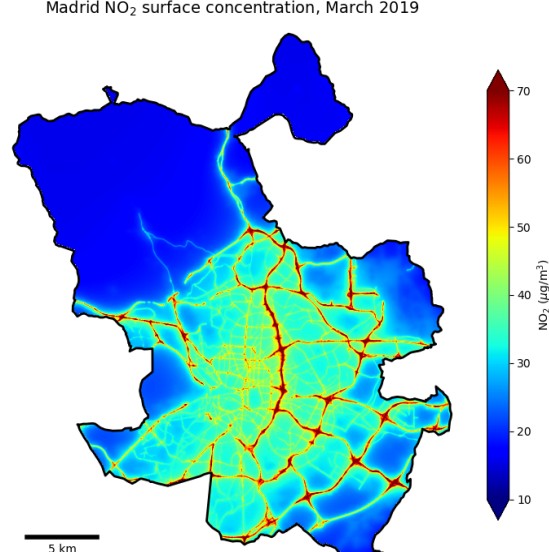

**Figure 10: Simulation of NO₂ surface concentrations by Retina averaged for March 2019, using TROPOMI observations for daily estimates of emission factors.**

The hourly $NO_2$ concentrations are validated against time series at the 24 sites of the reference network. The statistics of the hourly time series of Retina+TROPOMI and the regional CAMS ensemble are listed in Table 2. We use three different statistical parameters for evaluation:

- **Correlation**: the Pearson correlation coefficient. A value closer to 1 indicates a better capacity to describe the hourly dynamics of $NO_2$ concentrations. However, the model might still be off by a bias and/or a multiplication factor.

- **Bias**: difference between the average simulated value and the average observed value in the considered period. A negative value indicates a systematic underestimation of the simulation, while a positive value indicates a systematic overestimation of the simulation.

- **RMSE**: the root mean square error. A lower value indicates a smaller distribution of simulations around the true values. Note that the RMSE can be dominated by a bias.

As expected, the interpolated CAMS reanalysis shows a strong negative bias (-10.0 μg/m³ on average), particularly at roadside locations. The correlation, in contrast, is quite high (0.86 averaged over all 24 reference sites). This can be attributed to the CAMS ensemble members assimilating observations from a selection of background stations, namely ES0124, ES0126, ES1532, ES1939, ES1942, ES1947, and ES1946 (see CAMS, 2022).

Dispersion modelling based on TROPOMI-estimated emissions produces realistic ground concentrations: the absolute biases are reduced at most reference sites (from 10.0 to 0.8 μg/m³ on average), particularly at roadside and urban background





locations. However, this improvement comes at the cost of introducing more scatter, resulting in lower correlation (0.74 on

average), and an RMSE increasing from 18.1 to 19.3 μg/m$^3$.

**Table 2: Retina validation statistics for hourly surface NO$_2$ concentrations when optimising emissions using TROPOMI data (validation against interpolated CAMS concentrations in parentheses)**

| Location and type | Overpass hours (12-14 UTC) | | | | | | All hours | | | | | |
|---|---|---|---|---|---|---|---|---|---|---|---|---|
| | correlation | | RMSE (μg/m$^3$) | | bias (μg/m$^3$) | | correlation | | RMSE (μg/m$^3$) | | bias (μg/m$^3$) | |
| ES0115 street | 0.823 | (0.825) | 13.8 | (19.2) | -10.8 | (-17.3) | 0.777 | (0.868) | 18.7 | (21.9) | -5.9 | (-16.6) |
| ES0118 street | 0.610 | (0.794) | 20.4 | (32.8) | -15.7 | (-31.3) | 0.714 | (0.831) | 24.1 | (33.3) | -9.3 | (-28.7) |
| ES0120 street | 0.860 | (0.917) | 14.9 | (25.7) | -7.6 | (-21.8) | 0.747 | (0.814) | 19.9 | (19.8) | 5.6 | (-12.4) |
| ES1426 street | 0.848 | (0.959) | 9.7 | (12.2) | 4.8 | (-10.1) | 0.824 | (0.911) | 17.5 | (14.8) | 6.5 | (-8.7) |
| ES1521 street | 0.829 | (0.922) | 8.9 | (13.9) | -2.0 | (-11.5) | 0.738 | (0.900) | 19.1 | (21.0) | -0.5 | (-14.6) |
| ES1525 street | 0.700 | (0.851) | 16.9 | (17.4) | 10.9 | (-14.9) | 0.708 | (0.850) | 25.5 | (18.6) | 13.3 | (-12.1) |
| ES1938 street | 0.813 | (0.890) | 12.9 | (17.3) | -4.1 | (-12.8) | 0.713 | (0.850) | 20.5 | (15.9) | 5.0 | (-8.0) |
| ES1940 street | 0.794 | (0.865) | 12.5 | (20.5) | -4.4 | (-16.8) | 0.736 | (0.831) | 19.3 | (19.1) | 4.3 | (-12.6) |
| ES1943 street | 0.636 | (0.775) | 15.4 | (28.6) | 2.1 | (-26.4) | 0.722 | (0.823) | 26.8 | (38.4) | -5.2 | (-31.2) |
| ES0124 urban background | 0.836 | (0.961) | 10.4 | (13.4) | -2.2 | (-10.0) | 0.795 | (0.918) | 16.4 | (15.4) | 0.8 | (-9.6) |
| ES0125 urban background | 0.881 | (0.938) | 6.9 | (7.0) | -2.5 | (-4.9) | 0.817 | (0.935) | 23.1 | (22.1) | -11.4 | (-14.7) |
| ES0126 urban background | 0.906 | (0.962) | 8.5 | (7.8) | -4.3 | (-5.5) | 0.802 | (0.910) | 15.7 | (12.5) | -4.1 | ( -6.1) |
| ES1422 urban background | 0.746 | (0.810) | 11.2 | (12.6) | -5.9 | (-9.1) | 0.683 | (0.860) | 18.1 | (12.7) | 1.6 | ( -5.7) |
| ES1532 urban background | 0.849 | (0.937) | 8.0 | (8.2) | -2.9 | (-6.3) | 0.779 | (0.916) | 17.0 | (13.1) | -1.2 | ( -6.8) |
| ES1645 urban background | 0.777 | (0.877) | 10.1 | (10.2) | 1.0 | (-5.4) | 0.802 | (0.868) | 16.5 | (20.1) | -2.0 | (-12.4) |
| ES1937 urban background | 0.794 | (0.887) | 9.6 | (9.1) | -2.8 | (-5.6) | 0.711 | (0.893) | 19.3 | (13.5) | -0.9 | ( -5.9) |
| ES1939 urban background | 0.893 | (0.958) | 10.6 | (8.7) | -2.1 | (-3.2) | 0.586 | (0.807) | 21.5 | (12.7) | 7.2 | (3.1) |
| ES1941 urban background | 0.786 | (0.822) | 6.9 | (8.4) | -3.0 | (-6.1) | 0.744 | (0.867) | 18.9 | (16.9) | -6.0 | (-10.4) |
| ES1942 urban background | 0.663 | (0.937) | 11.7 | (11.2) | 2.1 | (-8.2) | 0.728 | (0.895) | 21.9 | (26.3) | -4.8 | (-19.7) |
| ES1944 urban background | 0.876 | (0.935) | 8.8 | (12.6) | -3.0 | (-9.5) | 0.782 | (0.872) | 16.6 | (17.2) | 2.2 | ( -9.3) |
| ES1947 urban background | 0.851 | (0.905) | 7.7 | (5.7) | 4.2 | (-0.8) | 0.740 | (0.854) | 17.9 | (14.9) | 5.8 | ( -4.1) |
| ES1193 suburban backgr. | 0.912 | (0.913) | 8.1 | (6.6) | -2.2 | (0.2) | 0.669 | (0.782) | 15.3 | (13.7) | 3.1 | (5.9) |
| ES1945 suburban backgr. | 0.820 | (0.862) | 7.2 | (6.2) | 3.9 | (-3.4) | 0.702 | (0.623) | 16.1 | (10.1) | 10.1 | (1.4) |
| ES1946 suburban backgr. | 0.852 | (0.968) | 9.6 | (7.3) | 1.8 | (-1.6) | 0.737 | (0.889) | 17.8 | (11.4) | 6.1 | (-0.5) |
| **Mean** | **0.806** | **(0.895)** | **10.9** | **(13.4)** | **-1.9** | **(-10.1)** | **0.740** | **(0.857)** | **19.3** | **(18.1)** | **0.8** | **(-10.0)** |

Satellite observations can only estimate emissions around overpass time. Wrong assumptions in the diurnal cycle for other

hours introduce an additional error. Therefore, the statistics improve when evaluated for overpass hours only (12:00 to 14:00

UTC): the correlation is higher (0.81) and RMSE is lower (10.9 μg/m$^3$).

A large negative bias is still found at ES0118 (Escuelas Aguirre), which is close to a busy intersection. Retina might

underestimate the local traffic flow here, or the additional pollution burden due to deceleration and acceleration of congested





traffic. A notable overestimation of NO$_2$ concentrations occurs at location ES1525 (Cuatro Caminos). This is subject to further investigation; it might be related to an overestimation of local traffic intensities.

## 3.2 Model accuracy under different network configurations

The Retina algorithm can also ingest in-situ measurements of one or more ground stations to estimate the emissions factors. The extent and the distribution of this observation network will affect the quality of the emission estimations and therefore

the accuracy of the NO$_2$ simulations.

First, we assess the influence of station location on emission optimisation using a single reference station, both with and without satellite data. Table 3 summarises the average validation statistics for hourly simulated NO$_2$ concentrations across all 24 observation locations.

**Table 3: Mean validation statistics using different reference stations for emission optimisation, sorted from high to low bias. The right column lists the corresponding total emission estimation. Outside parentheses: including TROPOMI data; inside parenthesis: excluding TROPOMI.**

| observations from | correlation | | RMSE (µg/m$^3$) | | bias (µg/m$^3$) | | emission (Mg NO) | |
|---|---|---|---|---|---|---|---|---|
| ES1943 | 0.731 | (0.723) | 26.7 | (29.0) | 12.8 | (17.2) | 2084 | (2638) |
| ES0125 | 0.748 | (0.758) | 24.7 | (24.5) | 12.3 | (13.2) | 2014 | (2200) |
| ES0118 | 0.739 | (0.750) | 23.7 | (24.1) | 9.7 | (12.6) | 2285 | (2755) |
| ES1942 | 0.759 | (0.760) | 21.1 | (21.1) | 7.9 | (7.9) | 1647 | (1651) |
| ES1941 | 0.753 | (0.755) | 21.5 | (21.1) | 8.2 | (7.7) | 1708 | (1669) |
| ES0115 | 0.752 | (0.756) | 20.4 | (20.6) | 6.0 | (7.6) | 1791 | (2086) |
| ES1521 | 0.781 | (0.784) | 18.6 | (18.3) | 3.8 | (3.6) | 1475 | (1496) |
| ES1532 | 0.775 | (0.782) | 18.3 | (17.8) | 2.3 | (2.1) | 1325 | (1344) |
| ES0126 | 0.764 | (0.778) | 18.4 | (17.7) | 1.4 | (1.1) | 1234 | (1244) |
| ES1937 | 0.776 | (0.781) | 17.8 | (17.6) | 0.7 | (1.0) | 1221 | (1260) |
| ES1645 | 0.772 | (0.778) | 18.1 | (17.7) | 1.5 | (0.7) | 1208 | (1144) |
| ES0124 | 0.783 | (0.793) | 17.7 | (17.1) | 1.0 | (0.4) | 1276 | (1263) |
| ES1422 | 0.768 | (0.768) | 17.7 | (17.8) | -1.4 | (-0.6) | 1217 | (1335) |
| ES1944 | 0.773 | (0.775) | 17.8 | (17.7) | -0.6 | (-1.0) | 1151 | (1144) |
| ES1940 | 0.776 | (0.778) | 17.7 | (17.6) | -1.8 | (-1.5) | 1103 | (1189) |
| ES1426 | 0.790 | (0.791) | 17.2 | (17.1) | -2.3 | (-2.3) | 1051 | (1060) |
| ES1938 | 0.772 | (0.765) | 17.9 | (18.0) | -1.7 | (-2.4) | 1172 | (1217) |




| | | | | | | | | |
|---|---|---|---|---|---|---|---|---|
| ES0120 | 0.781 | (0.778) | 17.5 | (17.4) | -3.2 | (-2.6) | 1083 | (1230) |
| ES1525 | 0.792 | (0.794) | 17.2 | (17.1) | -3.6 | (-3.6) | 1043 | (1059) |
| ES1939 | 0.792 | (0.796) | 18.7 | (18.9) | -8.2 | (-8.7) | 825 | (810) |
| ES1946 | 0.801 | (0.797) | 18.6 | (19.1) | -8.6 | (-9.5) | 631 | (531) |
| ES1193 | 0.720 | (0.722) | 21.0 | (21.3) | -9.4 | (-10.1) | 606 | (591) |
| ES1947 | 0.773 | (0.785) | 19.2 | (21.1) | -8.5 | (-12.6) | 667 | (377) |
| ES1945 | 0.687 | (0.741) | 26.1 | (26.0) | -16.9 | (-17.6) | 275 | (180) |
| TROPOMI-only | 0.740 | | 19.3 | | 0.8 | | 1185 | |
| 5 stations | 0.792 | (0.793) | 17.1 | (17.0) | -1.8 | (-1.9) | 1084 | (1075) |
| All stations | 0.791 | (0.792) | 17.3 | (17.2) | 1.2 | (1.2) | 1332 | (1336) |

The emission optimisation depends on the selected location. Local model issues or biased reference measurement can lead to

biases at other locations. For example, using data of road station ES1943 (Plaza Elíptica) introduces an overall bias of 17.2 µg/m³. Retina apparently underestimates the high concentrations at this site, and compensates by adding emissions, resulting in overestimations elsewhere. Combining the data with satellite data reduces the bias significantly. This improvement can be seen at all sites with large absolute bias. The impact of the satellite measurements remains limited, however, as only corrections around overpass time can be provided.

For 13 stations the mean absolute bias is below 4 µg/m³. At these sites Retina realistically describes the local concentrations with the provided traffic and residential proxies. Using one of these stations for emission optimisation, the RMSE and correlations are better than when using TROPOMI measurements alone, as the in-situ data provide valuable information on the entire diurnal emission cycle. As can be seen from the table, adding satellite data to these ground data does not improve the results significantly.

Increasing the number of in-situ stations enhances the accuracy of simulation results due to compensating errors, especially when the stations represent a balanced mix of street and background locations. The table shows the results of a test with 5 stations: ES1525, ES1940 (both street stations); ES0126, ES1937 (both background stations); and ES1947 (a suburban background station). It can also be seen that the effect of adding TROPOMI data becomes negligible, as the amount of valid daily satellite retrievals is small compared to the 120 daily measurements made by the 5 stations.

A final test incorporates all 24 reference stations for emission optimisation. No significant change in validation statistics is observed compared to the 5-station scenario. The resulting RMSE, 17.2 µg/m³, can thus be regarded as the systematic error inherent in the Retina approach for hourly NO₂ simulation.

Compared to this full in-situ analysis, TROPOMI-based emissions tend to attribute more emissions to traffic and less to residential activity (see Fig. S9), resulting in up to 5 µg/m³ higher concentrations on roads and up to 1.5 µg/m³ lower

concentrations in urban backgrounds.





### 3.3 Results of spatial concentration assimilation

Spatially assimilating the in-situ observations as described in Sect. 0 reduces simulation biases in the vicinity of an observation location (e.g. due to wrong local emissions), while at longer distances it reduces simulation errors due to inaccuracies in hourly background concentrations. Table 4 compares the validation results before (i.e. the plain dispersion simulation) and after the spatial concentration assimilation of reference measurements. We use a leave-one-out cross-validation: at each validation location the observations of the other 23 locations are assimilated in the simulation fields. Based on the average $NO_2$ concentrations (36.2 µg/m³) and the average RMSE found in the validation (17.2 µg/m³), the relative error for simulation of hourly $NO_2$ concentrations is estimated to be 48%. The data assimilation increases the correlation from 0.79 to 0.90 and reduces the RMSE to 13.0 µg/m³. Therefore, the relative error improves to 36% after assimilation, with local systematic biases remaining as the primary source of error.

**Table 4: Leave-one-out cross validation statistics for spatial concentration assimilation using all reference stations. (Statistics for model simulation, based on 24-station emission optimisation, inside parentheses.)**

| validation location and type | nearest | d [a] | n [b] | obs [c,d] | correlation | | RMSE [d] | | bias [d] | |
|---|---|---|---|---|---|---|---|---|---|---|
| ES0115 street | ES1422 | 0.9 | 738 | 44.6 | 0.882 | (0.829) | 16.2 | (16.3) | -9.7 | (-5.6) |
| ES0118 street | ES1939 | 0.8 | 733 | 57.1 | 0.876 | (0.761) | 24.7 | (22.3) | -20.3 | (-10.2) |
| ES0120 street | ES1938 | 1.7 | 736 | 37.5 | 0.930 | (0.792) | 9.7 | (17.7) | 0.6 | (5.4) |
| ES1426 street | ES1532 | 2.3 | 739 | 37.7 | 0.950 | (0.862) | 12.1 | (15.4) | 8.1 | (6.3) |
| ES1521 street | ES1940 | 2.4 | 730 | 37.1 | 0.923 | (0.807) | 14.4 | (16.3) | -8.5 | (-0.2) |
| ES1525 street | ES1938 | 1.6 | 737 | 38.0 | 0.858 | (0.768) | 19.5 | (22.0) | 11.7 | (11.7) |
| ES1938 street | ES1525 | 1.6 | 738 | 34.6 | 0.924 | (0.783) | 9.8 | (17.7) | 1.3 | (5.6) |
| ES1940 street | ES0120 | 1.8 | 739 | 36.6 | 0.936 | (0.787) | 9.3 | (17.0) | -0.9 | (4.5) |
| ES1943 street | ES0126 | 1.6 | 736 | 61.1 | 0.880 | (0.759) | 18.3 | (24.9) | -1.1 | (-7.8) |
| ES0124 urban background | ES0120 | 3.5 | 734 | 35.5 | 0.955 | (0.835) | 9.1 | (14.7) | -4.1 | (2.0) |
| ES0125 urban background | ES1943 | 4.2 | 739 | 43.3 | 0.923 | (0.847) | 17.0 | (21.4) | -8.5 | (-9.9) |
| ES0126 urban background | ES1943 | 1.6 | 738 | 35.7 | 0.945 | (0.842) | 8.6 | (13.9) | -2.4 | (-2.6) |
| ES1422 urban background | ES0115 | 0.9 | 668 | 34.0 | 0.928 | (0.785) | 8.6 | (14.9) | 2.3 | (2.7) |
| ES1532 urban background | ES1426 | 2.3 | 739 | 36.3 | 0.941 | (0.846) | 9.2 | (14.1) | -2.1 | (-0.1) |
| ES1645 urban background | ES1942 | 1.6 | 737 | 35.7 | 0.932 | (0.831) | 9.9 | (15.3) | -0.7 | (-1.3) |
| ES1937 urban background | ES1939 | 1.8 | 739 | 36.4 | 0.913 | (0.800) | 11.9 | (15.9) | -4.7 | (0.1) |
| ES1939 urban background | ES0118 | 0.8 | 738 | 25.9 | 0.870 | (0.693) | 15.1 | (18.8) | 9.9 | (7.8) |
| ES1941 urban background | ES1532 | 3.7 | 737 | 38.5 | 0.890 | (0.801) | 13.3 | (16.6) | -5.8 | (-5.1) |
| ES1942 urban background | ES1645 | 1.6 | 734 | 44.1 | 0.922 | (0.789) | 14.5 | (19.5) | -8.3 | (-5.2) |





| | | | | | | | | | | |
|---|---|---|---|---|---|---|---|---|---|---|
| ES1944 | urban background | ES1947 | 2.6 | 737 | 31.2 | 0.923 | (0.818) | 10.4 | (15.1) | -3.4 | (2.8) |
| ES1947 | urban background | ES1944 | 2.6 | 737 | 25.1 | 0.902 | (0.796) | 11.4 | (16.4) | 3.9 | (6.8) |
| ES1193 | suburban backgr. | ES0115 | 3.0 | 735 | 21.6 | 0.807 | (0.716) | 11.9 | (14.2) | 3.6 | (3.4) |
| ES1945 | suburban backgr. | ES1521 | 6.9 | 739 | 15.4 | 0.674 | (0.688) | 15.9 | (16.2) | 9.3 | (10.3) |
| ES1946 | suburban backgr. | ES1942 | 2.4 | 739 | 25.3 | 0.913 | (0.767) | 12.2 | (17.0) | 7.1 | (6.8) |
| **Mean** | | | | **734** | **36.2** | **0.900** | **(0.792)** | **13.0** | **(17.2)** | **-0.9** | **(1.2)** |


(a) Distance to the nearest observation site (in km)

(b) Number of measurements

(c) Mean observation at this location

(d) In units of $\mu g/m^3$

Note that an additional bias can be introduced at places where the covariance is wrongly defined. This happens for instance at street location ES0118 and city park ES1939 (El Retiro), which are at 800 m distance. The high concentrations found at ES0118 influence the spatial interpolation towards the urban background. Vice versa, the low concentrations measured in El Retiro park propagate towards the nearby street site, contributing to a negative bias.

Figure 11 illustrates the performance of Retina at a street location, an urban background location, and a suburban
background location. It shows the hourly $NO_2$ series for a representative week when only satellite observations are used in the simulation, representing the case for a city without any air quality observation network. This is compared with time series where all in-situ observations are used, representing a city with an extensive ground network. Not surprisingly, the best results are obtained when all reference measurements are used.



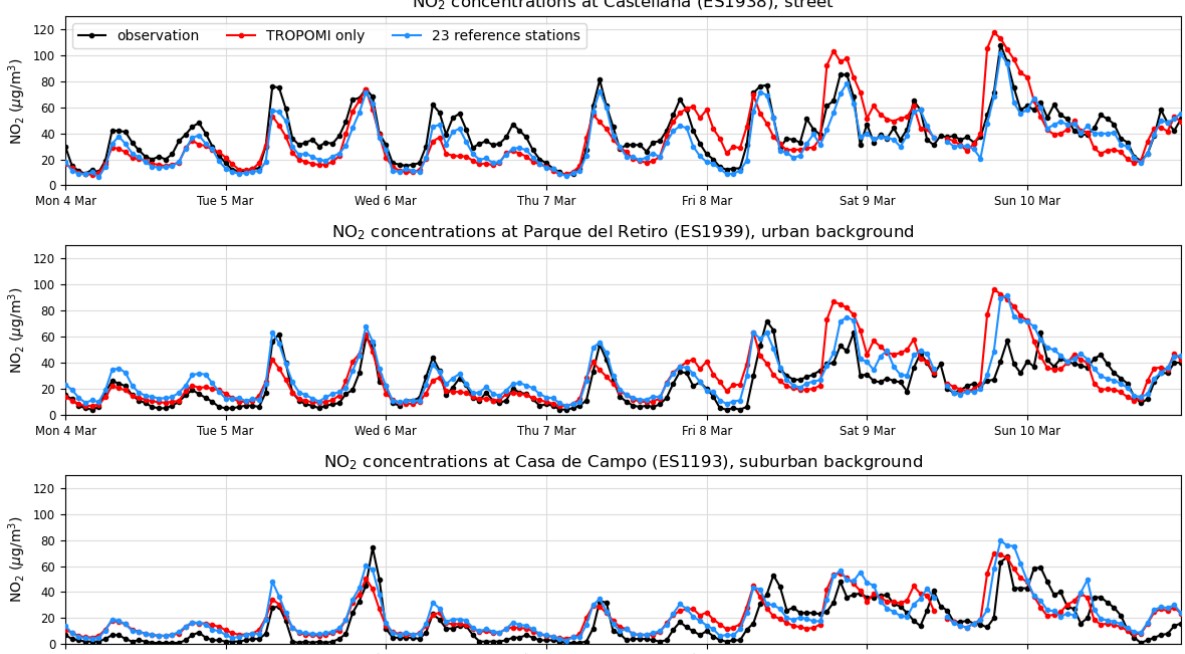

**Figure 11: Comparison of NO₂ time series at different locations for a week in March 2019. The red line represents simulations from the Retina algorithm using only TROPOMI observations for emission optimisation; the blue line represents the leave-one-out time series using data from all other reference stations.**

Figure 12 shows the average NO₂ concentration map of Madrid based on all hourly concentration fields of March 2019. Highest concentrations are found on and near the highways, such as the M30 surrounding the city centre in the East and South. Lowest concentrations are found in the sparsely populated El Pardo area in the north. Local concentration reductions are found in e.g. El Retiro park. Note the accumulation of air pollution in the southwest area of the municipality due to predominant winds (1-3 Beaufort) from the northeast in this period.





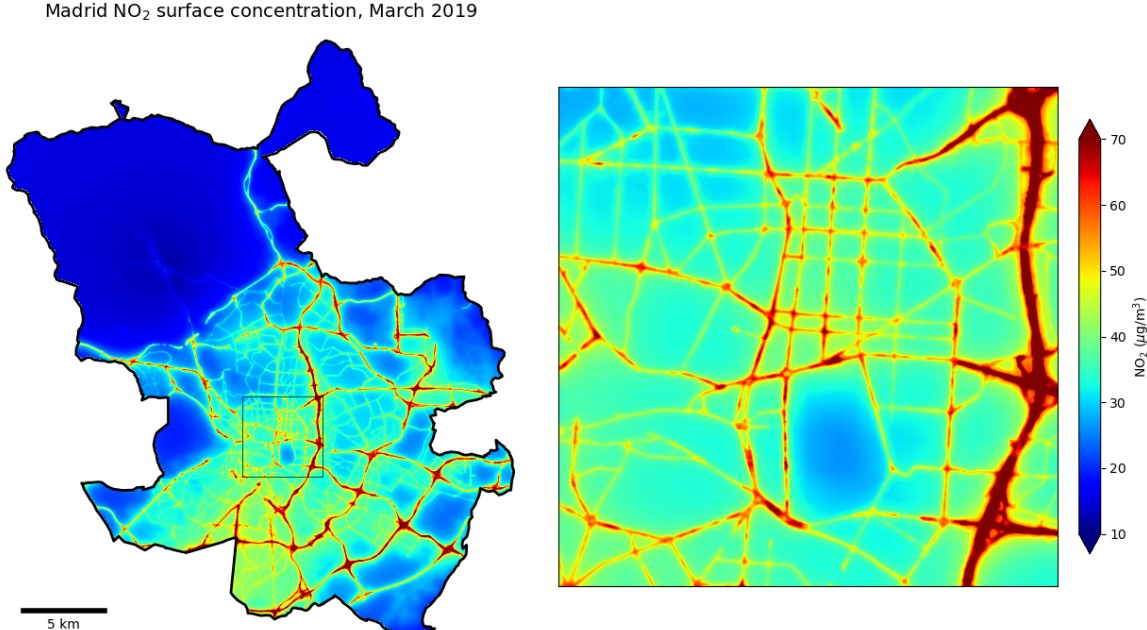

**Figure 12: Average surface NO$_2$ concentrations in Madrid for March 2019, using the Retina algorithm with hourly data from 24 reference stations for emission estimation and spatial assimilation. The right panel zooms in on an 5x5 km$^2$ area around Retiro city park, where concentrations are notably lower than in nearby roads and built-up areas.**

## 4 Discussion

Retina v2 introduces a more realistic NO$_2$ chemistry scheme, an improved background concentration estimation, and a better stabilised emission optimisation. The spatial assimilation of measurements is improved by including time-dependent dispersion characteristics in the model error covariances. Most notably, the algorithm is now capable to incorporate tropospheric column concentrations retrieved with satellite instruments, such as TROPOMI on the Sentinel-5 Precursor (S5P) satellite.

Satellites in polar orbits, such as S5P, pass over the same area just once each day, therefore missing a substantial part of the diurnal cycle of NO$_2$. Direct assimilation of NO$_2$ satellite observations is not very useful due to the relatively short lifetime of NO$_2$, which limits the system's memory to just a few hours. Instead, we use the satellite retrievals to improve estimations of urban NOx emissions. As the number of daily TROPOMI observations over the urban area is limited (14 on average for the Madrid domain), it is important to get the most out of each satellite retrieval by interpolating the model simulations to the individual footprints at exact overpass time. Applying the averaging kernel minimizes errors resulting from profile assumptions in the retrieval method.





## 4.1 Comparison with relevant studies


$NO_2$ concentrations in urban areas vary strongly in space and time. Unsurprisingly, the CAMS regional ensemble is unfit to represent local $NO_2$ concentrations in urban areas. Due to its coarse resolution, its interpolated values underestimates concentrations by 10.0 µg/m³ in Madrid in March 2019. However, the CAMS ensemble provides valuable input data for background concentration estimation and $NO_2$/NOx column ratios for downscaling algorithms such as Retina.

The validation results show that an urban dispersion model can successfully be built based on CAMS input data and proxy data for traffic and residential emissions. Validation of the hourly $NO_2$ simulations based on periodic emission optimisation by TROPOMI show a reduction of the mean bias to 0.8 µg/m³ and an average RMSE of 19.3 µg/m³. Part of the error is caused by wrong assumptions in the diurnal emission cycle, as TROPOMI is only able to capture the emissions around its overpass time. Finding a better *a priori* diurnal emission cycle is subject to further investigation.

CALIOPE-Urban, an advanced dispersion model based on a detailed emission inventory and running on a supercomputer, produces an RMSE of 23 µg/m³ for hourly simulations in 2019 for the city of Barcelona (Criado et al., 2023). This RMSE reduces to 16 µg/m³ when reference data of 12 stations is spatially assimilated using Universal Kriging, and further to 12 µg/m³ when also an additional basemap layer based on Palmes-tube measurements is included. Schneider et al. (2017) find a citywide RMSE of 14.3 µg/m³ for a similar data fusion method of 24 low-cost sensors in the EPISODE model for Oslo in

January 2016. From Table 4 can be seen that these figures are comparable to the RMSE of 13 µg/m³ by Retina when all reference measurements are spatially assimilated.

Alternatively, several studies use a machine learning approach to generate hourly surface concentrations maps from a collection of data sets. Kim et al. (2021) train a predictive model including data from TROPOMI and 340 reference stations in Switzerland and northern Italy, resulting in a spatio-temporal correlation of 0.79 with 40 test sites. Table 3 and 4 show that

the correlation of Retina simulations with reference measurements in Madrid is 0.74 when only TROPOMI is used (i.e. no surface measurements), increasing to 0.79 when 5 or more surface stations are also used for emission optimisation. Best correlation (0.90) is obtained when the reference measurements are also spatially assimilated.

Fu et al. (2023) use data from 266 reference stations and 666 low-cost sensors (LCS) in the Tangshan area (East China). TROPOMI data are used in XGBoost models to fill in missing data at reference sites and to enhance the observations at LCS

sites. When trained with reference data only, their predictive model has a correlation of 0.79 and an RMSE of 17.1 µg/m³. The RMSE improves to 16.9 µg/m³ when including TROPOMI, and further to 16.3 µg/m³ when all LCS observations are also included in the training. Table 3 shows that for Retina-Madrid the RMSE is 17.0 µg/m³ when 5 reference stations for emission optimisation are used.

By adding more in-situ data, the RMSE of hourly simulations remains around 17 µg/m³, corresponding to a relative error of

48%, which can be considered as the systematic error of the Retina dispersion modelling. This is an improvement over the previous version described in Mijling (2020), which had an estimated error of 58%. More research is needed to further reduce this error by addressing its various sources, such as:



- Using better proxy data, particularly regarding the relative distribution of traffic volumes.
- Including emission hot spots from industry and power generation.
- Improving local dispersion modelling by accounting for e.g. traffic junctions, speed bumps, and street canyons.
- Using more realistic estimates of the background concentration field.
- Improving NOx chemistry, e.g. by introducing variable $NO_2$ lifetimes.

Urban NOx emissions can be calculated from the emission optimisation results by summing Eq. (2) over hours and grid cells. The observation-based NOx emission estimates for March 2019 in Madrid vary between 1185 Mg NO (when derived from TROPOMI observations) and 1336 Mg NO (when derived from in-situ observations at 24 locations). This is larger than the 705 Mg NO found in the CAMS emission inventory for this month (Soulie et al., 2024) but is in correspondence with the DECSO v6.3 inventory (Van der A et al., 2024; based on TROPOMI observations and chemical transport model calculations) being a factor 2 larger than the CAMS inventory. The potential of Retina to estimate realistic urban emissions is subject of further investigation.

## 4.2 Calculation time

The Retina v2 algorithm is implemented in Python scripts. Calculations were performed on a Linux workstation with an Intel Core i7-9700 at 3GHz, having a single CPU and 8 cores. The total calculation for a high-resolution surface simulation at a certain hour takes 75 s. The dispersion kernel calculation using AERMET and AERMOD takes 3.4 s of this time. The preparation of the emission proxy data takes 6.6 s, mostly spent on interpolation and gridding of the traffic volumes. The surface concentration simulation takes 60.5 s, of which 96% of the time is spent on Hadamard product calculations. Finally, the spatial interpolation of the surface measurements takes 4.2 s.

The emission optimisation is repeated once every simulation day. This takes 188 seconds if only surface measurements are considered (159 s are spent in emission proxy preparation). It takes ~75 s longer if TROPOMI observations are also taken into account; time which is needed to perform the column simulations and the spatio-temporal interpolation to individual retrieval footprints.

In practice the computational time is less since the emission proxy calculations are shared between the simulation and the emission optimisation, requiring computation only once. The Hadamard product calculation scales linearly with the number of receptor points. For the surface concentration simulation this forms currently a computational bottleneck. However, as it involves straightforward matrix operations, the total computation time can be significantly reduced by parallelizing these tasks across multiple cores or GPUs.

## 4.3 Use of low-cost sensor data

As shown before in Mijling (2020), the Retina algorithm can also be applied to networks of low-cost sensors. Based on Bayesian principles in both emission optimisation and spatial assimilation, it effectively manages the greater inaccuracies associated with LCS data. The larger errors in the in-situ observations will slow down convergence to practical emission





estimates in the optimisation phase (i.e., a longer lag period), but this is not necessarily a problem when emission trends are small over time.

Note that most $NO_2$ low-cost sensors used in current experimental networks suffer from creeping biases (Li et al., 2021, WMO, 2023). Also reference measurements can be biased due to interfering gases (Steinbacher et al., 2007) or poor maintenance. Although always special care must be taken to remove this bias before application in Retina, integrating satellite measurements can help to reduce the introduced bias in the monitoring system.

## 5 Conclusion

The Retina algorithm has been designed to produce realistic high-spatiotemporal-resolution maps of urban air pollution based on heterogeneous air quality measurements. In this study, we implemented the updated Retina algorithm for $NO_2$ concentrations in Madrid and assessed the performance under different observation scenarios during March 2019. The updated algorithm, Retina v2, is faster and more accurate than its predecessor described in Mijling (2000). Most notably, it is now capable to incorporate tropospheric column concentrations retrieved with satellite instruments.

The use of proxy data for the description of urban emissions allows for convenient portability to other urban domains. Periodic emission optimisation guarantees that simulations match the observations, either satellite measurements, in-situ measurements, or both. Physics-based and running with modest computational power, Retina has comparable or better performance than data fusion methods depending on advanced, computational-demanding dispersion models, as well as machine learning approaches depending on extensive of datasets.

When emissions are optimised using only TROPOMI measurements (representing the case of a city without an in-situ network), simulations of hourly $NO_2$ concentrations in March 2019 show a citywide RMSE of 19.3 µg/m³ with a bias of 0.8 µg/m³. More accurate results are achieved when hourly in-situ measurements are used, as they allow for a better estimation of the diurnal emission cycle. However, if only a single station is available, and its measurements are biased or located in an area where dispersion modelling is problematic—due to e.g. incorrect proxy data—it can introduce systematic biases across the entire model domain. Incorporating satellite measurements or data from additional ground stations helps to reduce this bias.

The spatial interpolation of in-situ measurements in the simulation results improves the accuracy significantly: near observation sites it reduces simulation biases (e.g. due to inaccurate local emissions), and over larger distances it reduces simulation errors due to errors in background concentrations. Generally, including more stations leads to better results. Using all 24 ground stations in Madrid, the average correlation of hourly $NO_2$ time series increases to 0.90, with an RMSE of 13.0 µg/m³, corresponding to a relative error of 36%. Occasionally, the spatial interpolation introduces an extra bias. This suggests that there is further room for improvement in the covariance model used for interpolation.

The assimilation experiments show that the added value of TROPOMI measurements becomes negligible when hourly data from 5 or more stations at representative locations is included. For many cities, however, TROPOMI can made a significant

contribution. From the approximately 2800 European cities with a population above 50,000, the European Environment Agency (EEA) AirBase (EEA, 2018) lists 2035 cities with at least one $NO_2$ monitoring station, and only 71 cities with 5 or more $NO_2$ stations (see Table S1).

The impact of satellite measurements in the Retina algorithm will be larger if observations at different times throughout the day could be included. Therefore, the next step will be preparation for data of the Sentinel-4 instrument aboard the MTG-S satellite, expected to be launched in late 2025. Operating from a geostationary orbit, Sentinel-4 will provide hourly measurements of $NO_2$ in Europe at 8x8 $km^2$ resolution with a revisit time of approximately 60 minutes. Once alternatives for CAMS background concentrations are available beyond the European domain, applications can extend to geostationary

instruments such as GEMS (aboard GEO-KOMPSAT-2B, monitoring East Asia) and TEMPO (aboard Intelsat-40E, monitoring North America).

**Code and data availability**

The source code of the Retina v2 model and input data needed to reproduce the results in this study are publicly available via GitLab at https://doi.org/10.21944/retina-v2-madrid-2019 (Mijling, 2025). A list of online sources of raw input data can also

be found here.

**Supplement**

The supplement related to this article is available online at: *XXX*.

**Author contribution**

BM conceptualized and designed the Retina algorithm, including coding and data analysis, and wrote the draft of the

615 manuscript. HE provided scientific feedback and suggestions for algorithm improvement. PM and SH were involved in code improvement and postprocessing of the model output. DG and MdV were responsible for collecting the in-situ data. All co-authors helped in editing suggestions to the manuscript.

**Competing interests**

The authors declare that they have no conflict of interest.



## Acknowledgements

The authors wish to acknowledge the people behind the data sources used in this study, most notably the Madrid authorities (traffic and reference measurements), the CAMS community (background and column concentrations), and the TROPOMI team (tropospheric $NO_2$ column retrievals).

## Financial support

This research has been supported by the European Space Agency (ESA/ESRIN) in the CitySatAir project (contract no. 4000131513/20/I-DT), part of the Earth Observation for Society program.

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
