# Peer review of "High-resolution mapping of urban NO2 concentrations using Retina v2: a case study on data assimilation of surface and satellite observations in Madrid"

_EGUsphere, 2025_

## Author Comment (AC2)

**Response to Reviewer #1**

We thank the reviewer for the careful evaluation of our manuscript and for the constructive comments and suggestions. Our point-by-point responses are provided below, with reviewer comments shown in blue and our replies in black.

This paper presents a thorough update on previous work, highlighting several important improvements in the Retina algorithm. The topic is relevant, and the paper is both well-structured and clearly written. As the authors emphasize, the most significant advancement over their earlier work is the integration of satellite data into the data assimilation scheme, which apparently would only have an added value when less than 5 monitoring stations are available in a specific city.

Specific comments:

1. Since the main novelty of the presented methodology lies in the integration of satellite data (TROPOMI) for urban $NO_2$ modeling, a longer validation period would be highly valuable. The current one-month evaluation period may not fully capture the seasonal variability in satellite retrieval quality, atmospheric dynamics, and emissions. Statistical performance metrics may therefore exhibit seasonal dependence, potentially leading to less robust conclusions regarding the added value of satellite data.

To get a better insight in the seasonal behaviour of the Retina algorithm, we performed a processing for 2019 for three different cases:

    (A) Emission optimisation based on TROPOMI only
    (B) Emission optimisation based on 24 surface stations only
    (C) Emission optimisation and spatial assimilation of 24 surface stations

The table below shows the validation statistics per month, based on time series of hourly simulation, averaged over the 24 stations. March 2019, indicated in bold font and evaluated in the main text, offers a reasonable approximation for the yearly performance. This table is now included in Section S7 of the Supplemental Material, and references to the algorithm's seasonal behaviour are included in Sections 3.1 and 5.

| 2019 | Obs. (µg/m³) | Correlation | | | RMSE (µg/m³) | | | Bias (µg/m³) | | |
|---|---|---|---|---|---|---|---|---|---|---|
| | | A | B | C | A | B | C | A | B | C |
| January | 55.6 | 0.760 | 0.788 | 0.908 | 23.7 | 22.2 | 16.3 | -4.7 | -0.5 | -1.5 |
| February | 55.4 | 0.790 | 0.835 | 0.909 | 22.2 | 19.6 | 15.9 | -4.9 | -1.2 | -1.6 |
| **March** | **36.2** | **0.753** | **0.792** | **0.900** | **18.5** | **17.2** | **13.0** | **-2.7** | **1.1** | **-0.9** |
| April | 27.3 | 0.728 | 0.752 | 0.892 | 15.0 | 14.6 | 11.0 | -2.4 | 0.6 | -0.8 |
| May | 22.3 | 0.705 | 0.719 | 0.877 | 13.7 | 13.6 | 10.2 | -2.6 | -0.1 | -1.1 |
| June | 24.7 | 0.698 | 0.705 | 0.855 | 14.3 | 14.0 | 11.1 | -0.4 | 0.5 | -0.7 |
| July | 26.2 | 0.716 | 0.693 | 0.877 | 15.7 | 15.8 | 11.4 | -2.1 | -0.2 | -0.7 |
| August | 25.9 | 0.777 | 0.795 | 0.901 | 15.9 | 15.1 | 11.8 | -1.8 | -0.4 | -1.0 |

| | | | | | | | | | | |
|---|---|---|---|---|---|---|---|---|---|---|
| September | 31.5 | 0.741 | 0.795 | 0.901 | 18.5 | 15.8 | 12.6 | -0.3 | -0.4 | -1.3 |
| October | 41.4 | 0.753 | 0.794 | 0.890 | 19.6 | 17.5 | 14.1 | -2.9 | 0.0 | -1.1 |
| November | 27.4 | 0.836 | 0.844 | 0.925 | 12.2 | 11.5 | 8.5 | -2.2 | -0.0 | -0.6 |
| December | 40.1 | 0.826 | 0.836 | 0.925 | 15.5 | 14.4 | 10.4 | -3.1 | 0.1 | -0.7 |
| Average | 34.5 | 0.757 | 0.779 | 0.897 | 17.1 | 15.9 | 12.2 | -2.5 | -0.0 | -1.0 |

As shown in the table, $NO_2$ observations peak during winter months. This is due to lower mixing heights and colder temperatures (leading to stronger $NO_X$ emissions from e.g. heating and longer atmospheric lifetimes of $NO_2$). During the summer months, both cases (A) and (B) show the lowest RMSE, but also show poorer correlation. This can be explained the higher ratio of the RMSE to the mean observations of $NO_2$ during summer.

Note that the results for case (A) in March differ slightly from those in Table 3 (where for TROPOMI-only the city-wide correlation is 0.740, RMSE is 19.3 µg/m³, and bias is 0.8 µg/m³). This can be explained from the starting point of the processing (November 2018) beigin different from the main text (January 2019).

Additionally, in case (A) all months show negative biases, with the largest biases occurring in winter. This is likely due to the use of a fixed diurnal profile for residential emissions throughout the year (see Section S4). Introducing a seasonal component in this profile could improve the results.

2. The method used to estimate background $NO_2$ concentrations via a line integral over the municipal perimeter raises several questions:
a. Does $e_v$ represent the local wind direction? If so, at what altitude or vertical level is the wind taken from?

We evaluate the wind direction at 10 m altitude at the centre of the domain. This wind is taken homogeneous across the entire domain. (Note that this is the same wind which is used in the dispersion modelling.) This clarification has been added in Section 2.2.1.

b. Eq. 1 resembles a mass conservation approach, but it lacks a temporal term—how is accumulation of pollutants within the domain accounted for?

Please note that the line integral is not based on a mass conservation approach. Instead, it represents a partial integration over the CAMS concentrations found along the domain perimeter, finding a representative (uniform) background concentration which flows into the model domain. The vector $e_v$ is needed to (a) discriminate between concentrations flowing into the domain and out of the domain and (b) put more weight to line segments perpendicular to the wind direction.

c. Since the method depends on integrating along the perimeter, does this imply that the background concentration depends on the chosen perimeter?

In our approach the background concentration is a scalar added to the locally produced $NO_2$ field. Changing the domain (or perimeter) will indeed alter the estimated background

concentration. However, it also affects the calculated field of locally produced $NO_2$. As a result, the net effect remains relatively insensitive to the specific choice of domain boundary.

d.  In the special case where $\mathbf{e_v} \cdot \mathbf{n} > 0$ for the entire perimeter (i.e. all wind is outflow), the integral appears ill-posed. How is this handled in the analysis?

Since we assume a homogeneous wind across the entire domain rather than using local wind variations, this situation cannot occur.

e.  Is this a novel approach? If so, could the authors justify its use and provide a comparison with background concentrations derived from station data within the domain?

To our knowledge, this is a novel approach. It is motivated by the need for a straightforward method to estimate background concentrations using the coarse-resolution data from the CAMS regional ensemble, while avoiding double counting of $NO_2$ from local emission sources. Validation of this method is now discussed in Section S1 of the Supplemental Material, based on the figure below.

[Figure]

The top panel shows $NO_2$ measurements from two suburban background stations: ES1193 (Casa de Campo) and ES1945 (El Pardo), which consistently record the lowest concentrations in the area. A third station, ES1946, also classified as suburban background, is excluded due to its elevated readings, likely influenced by nearby urbanization and proximity to Barajas International Airport. The time series show that the lowest $NO_2$ concentrations alternate between the two selected stations. This variation is partly explained by wind direction, represented by black arrows indicating 6-hour intervals. Typically, El Pardo registers lower $NO_2$ levels when clean air arrives from the northeast to northwest, whereas Casa de Campo, being downwind, includes additional local pollution contributions.

The bottom panel compares the lowest $NO_2$ concentration measured between the two stations with the background concentration calculated from CAMS data along the partial municipal perimeter, as described in Section 2.2.1. The close agreement between the calculated background and the observed minima suggests that this method provides a realistic estimate of background $NO_2$ under varying meteorological conditions.

3. The manuscript suggests that the added value of TROPOMI measurements becomes negligible when data from 5 or more stations are available. However, this rule of thumb may not be sufficiently robust, as it oversimplifies the issue. Other factors (such as city size, $NO_2$ concentration levels, local meteorological conditions, …) can significantly influence this threshold.

We agree that the rule of thumb regarding the added value of TROPOMI in relation to ground stations is not universally applicable to other cities. Accordingly, we have revised the corresponding paragraph in the Conclusion to present a more nuanced statement:

"The assimilation experiments for Madrid indicate that the added value of TROPOMI $NO_2$ measurements becomes negligible when hourly data from five or more ground-based stations at representative locations is available. *However, this rule of thumb cannot be directly applied to other cities, as the contribution of TROPOMI depends on various factors, including city size, $NO_2$ concentration levels, and local meteorological conditions.* Nevertheless, in many urban areas—especially those with sparse in situ monitoring—TROPOMI has the potential to provide substantial added value. Among approximately 2800 European cities with a population over 50,000, the European Environment Agency's AirBase (EEA, 2018) lists 2035 cities with at least one $NO_2$ monitoring station, but only 71 cities with five or more $NO_2$ stations (see Table S1)."

4. The manuscript states that traffic flow between counting locations is estimated using inverse-distance weighting interpolation, applied separately for highways and primary roads. However, since traffic volumes can vary significantly over short distances, especially in complex urban settings, this method might lead to unrealistic flow patterns. Could the authors justify the use of this interpolation approach and provide information on how its performance was assessed? Specifically, has any cross-validation been performed (e.g., removing some sensors and comparing interpolated vs. observed counts)?

Indeed, we recognize that traffic volumes can vary significantly over short distances in urban environments, and that inverse-distance weighting (IDW) interpolation may not fully capture such local variability. Our choice of this method was driven by the need for a practical and computationally efficient approach that could be applied consistently across many road segments, given the spatial resolution and availability of traffic data.

To assess the performance of the interpolation, we conducted a leave-one-out cross-validation based on daily traffic volumes. This was done separately for highways and primary roads, in line with how the interpolation algorithm is applied in Retina. For highway locations (n = 390), the average observed traffic volume was 79.5 vehicles per minute, while for primary roads (n = 3073), it was 6.2 vehicles per minute.

The resulting scatter plots of observed versus interpolated daily traffic volumes are provided below. Performance metrics are summarized in the table, reported as error ranges in vehicles per minute. The correlation is relatively low, supporting the reviewer's concern and highlighting the limitations of the current approach. We have now noted this explicitly in Section 4.1, where we emphasize that improved representation of traffic emissions—especially methods that better account for the relative distribution of traffic volumes—would be a valuable enhancement and are a priority for future development.

[Figure]

| Error range (vehicles per minute) | Fraction of primary road locations within error range | Fraction of highway locations within error range |
|---|---|---|
| ±1 | 23.1 % | 8.5 % |
| ±2 | 41.9 % | 14.6 % |
| ±3 | 57.3 % | 19.7 % |
| ±4 | 70.2 % | 23.1 % |
| ±5 | 79.1 % | 26.2 % |
| ±6 | 85.7 % | 31.0 % |
| ±7 | 90.3 % | 37.4 % |
| ±8 | 92.8 % | 40.8 % |
| ±9 | 94.1 % | 44.4 % |
| ±10 | 95.5 % | 47.2 % |
| ±11 | | 51.8 % |
| ±12 | | 54.9 % |
| ±13 | | 57.4 % |
| ±14 | | 59.7 % |
| ±15 | | 62.8 % |
| ±16 | | 66.9 % |
| ±17 | | 69.0 % |
| ±18 | | 71.3 % |
| ±19 | | 72.3 % |
| ±20 | | 75.6 % |

5. Pg. 27 line 518: The manuscript compares the Retina model's performance in Madrid with that of Kim et al. (2021), who trained a model using TROPOMI and 340 reference stations in Switzerland and northern Italy, obtaining a similar spatio-temporal correlation (0.79). However, several important differences limit the validity of this comparison: (i) Kim et al.'s study covers a much longer period (June 2018 to May 2020), including winter months, when satellite data is more frequently missing due to cloud cover—especially in complex alpine orography, which also affects the satellite's ability to translate column densities into surface concentrations. (ii) Elevated regions like the Alps can introduce systematic biases in satellite-derived $NO_2$ due to vertical gradients in $NO_2$ distribution and reduced sensitivity near the surface. (iii) Additionally, the amount of stations is much higher in the Kim et al. study (340 stations vs. 24 in Madrid), making their results spatially and statistically more robust. I suggest the authors reconsider the framing of the comparison or add more nuance to highlight the limitations and contextual differences that affect model performance in each case.

We agree, and now better frame the comparison between Retina and the mentioned machine learning approaches:

"Alternatively, several studies use a machine learning approach to generate hourly surface concentrations maps from a collection of data sets. *While our study focuses specifically on the urban area, these approaches typically cover larger regions and incorporate a broader and more diverse set of in-situ measurement locations.*"

Minor comments:

1. pg 9 line 199: "See 0"
2. pg 17 line 360 "Sect. 0."
3. pg 21 line 447: correct "Sect. 0"

All broken references have been restored.

4. pg 26, the manuscript states that "Direct assimilation of $NO_2$ satellite observations is not very useful due to the relatively short lifetime of $NO_2$ [...]" I would suggest that the issue may not lie in the inherent utility of the data, but rather in how the data is adapted and integrated into the model.

We clarified our motivation for using TROPOMI data to estimate emissions, rather than directly updating concentration fields, by revising this sentence to: "As a result, directly assimilating $NO_2$ satellite observations into concentration fields has limited utility, given the short atmospheric lifetime of $NO_2$ which limits the system's memory to just a few hours."

---

## Author Comment (AC3)

**Response to Reviewer #2**

We thank the reviewer for the careful evaluation of our manuscript and for the constructive comments and suggestions. Our point-by-point responses are provided below, with reviewer comments shown in blue and our replies in black.

Dear authors,

I would like to thank you for a very interesting read! I have a couple of questions that I'd like you to reflect on, but overall I am very happy with the quality of the manuscript and the described research.

Eq.1 - Looking back at Figure 2, the assumption that $b$ can be assumed constant along the perimeter of the city seems a bit optimistic? There is a factor of six difference in the concentration along the border in the north and the south of the city in March 2019, which suggests that a westerly or easterly wind would cause a much higher flux across the border in the south than in the north.

We agree with the reviewer that the assumption of a constant background will not describe well the influx of $NO_2$ along the perimeter when nearby and upwind emission sources are unevenly distributed along the perimeter. However, Figure 2 might give a misleading impression of the distribution of background concentrations due to the spatial interpolation of the coarse CAMS grid and averaging over time. For example, Table 4 shows that the suburban background concentration in the north (at ES1945) is 15.4 µg/m³, while in the south (at ES1193) it is 21.6 µg/m³—a difference that is significant, but far smaller than the factor suggested by the reviewer. A validation of our method is now included as Section S1 in the Supplemental Material. Nevertheless, we agree that a location-dependent background field would provide a more realistic representation. This limitation is acknowledged in Section 4.1 and will be addressed in future versions of the algorithm.

And relatedly, you write (L179): "Other sectoral emission, e.g. from industry, will be accounted for indirectly in either an increased background field or in additional residential emissions." Such industrial sites are likely not equally distributed along the border, further increasing inhomogeneities in the background border flux. So my question is the following: Why not instead discretize the border along the $l$ and $z$, and apply the same dispersion kernel that is used inside the city?

This is an interesting suggestion. However, application of the dispersion kernel will not be straightforward. The dispersion kernel describes the evolution of a plume originating from a point source at a defined injection height. In contrast, the background concentration entering the domain is assumed to be vertically mixed within the boundary layer, representing a vertically uniform concentration column rather than a plume from a discrete source.

To apply the kernel in this context, we would need to represent this vertically mixed background column as a distribution of effective sources across both height and location—a transformation that would require assumptions about source strength, vertical injection profiles,

and transport history outside the domain. At present, this is beyond the scope of our implementation.

Nonetheless, we agree that more detailed spatial structuring of the boundary inflow is desirable, and we will explore this in future work, particularly in cases where background contributions are expected to be highly anisotropic or dominated by near-boundary sources.

L204 - Is it reasonable to assume that residential emissions are similar during weekdays and in the weekend?

We acknowledge that that residential emissions from activities like cooking and heating may vary between weekdays and weekends. However, quantifying this weekly cycle from literature is challenging due to limited data. For example, the CAMS-TEMPO emissions inventory (Guevara et al., 2021) does not include a weekly cycle for residential and commercial combustion. Moreover, the influence of weekly variability in residential emissions is likely small relative to other sources of model uncertainty, as residential emissions account for only 16% of total $NO_X$ emissions in the Madrid municipal area, according to the CAMS global inventory (Soulie et al., 2024). Therefore, slight differences between weekdays and weekends in this sector are unlikely to significantly impact the overall model results.

The Retina algorithm could provide further insight into the relevance of prescribing weekly residential emission profiles, but we feel that this lies beyond the scope of the current study. We added to Section 4.1: "Finding a better a priori diurnal *and weekly* emission cycle is subject to further investigation."

It is worth noting that the algorithm does capture *seasonal* variations in residential emissions (due to i.e. heating demand), which are more pronounced and impactful than weekly fluctuations.

L230-L234 - Can you explain a bit about how AERMOD treats dispersion through street canyons? If only one dispersion kernel is calculated for each combination of wind speed and direction, stability and boundary layer height, the model cannot deal with variations in the built environment or even in roughness length (which I imagine can vary a lot from the sparsely populated northern area to downtown Madrid), correct? Do you expect this to result in large errors?

We chose to use a uniform surface roughness length as a pragmatic compromise between computational efficiency and physical representativeness. The reviewer rightly points out it limits the model's ability to capture the complex flow and dispersion processes within street canyons, such as recirculation zones or pollutant entrapment.

We realize that this simplification introduces uncertainty at the street scale, particularly under low wind conditions or in deep canyons, and we have identified it as a source of model error in Section 4.1. We are planning to introduce a parametrized version of the street canyon effect (for instance, following Vardoulakis et al., 2003) in the next algorithm version. Combined with a more realistic traffic model we expect to substantially reduce the model uncertainty.

L269-L270 - The dispersion model calculates concentrations of NOx but rather than assimilating NOx measurements, you assimilate NO2 measurements. You get these from the

XGBoost algorithm, which introduces non-linearity to the system. While you explicitly mention that you ignore the dependence on O3 (L85-L86 of the SI), there is also the dependence on e.g. the temperature and the SEA (L264-L264). How do you reconcile this with the fact that a Kalman Filter assumes a linear measurement operator H?

The non-linearity introduced by ozone chemistry is accounted for in the calculation of the $NO_2/NO_X$ ratio, represented by $r_i$ in Equation (S1). The summation describes the contribution to the local $NO_X$ concentration by sector, and does not contain any ozone dependence. The ratio $r_i$ is evaluated with the XGBoost model based on local values of the predictors. It is assumed to remain approximately constant for small changes in NOx due to small perturbations in emission factors $x_j$. Consequently, Equation (S4) represents a local linearization of the model state. This linear approximation may become inaccurate if updates in emission factors are too large. However, since emission factors typically change slowly over time relative to the daily update frequency, the Kalman filter is expected to iteratively converge to stable and consistent emission estimates.

Minor comments

L199: "See 0?"

L575: Mijling (2000) should be Mijling (2020)

Mult: Please also check the manuscript for many different occurrences of Sec. 0.

Thank you for pointing this out. The references to sections and papers have been corrected.

**References**

Vardoulakis, S., Fisher, B.E., Pericleous, K. and Gonzalez-Flesca, N., 2003. Modelling air quality in street canyons: a review. *Atmospheric environment*, 37(2), pp.155-182.